# Novel secreted STPKLRR from *Vibrio splendidus* AJ01 promotes pathogen internalization via mediating tropomodulin phosphorylation dependent cytoskeleton rearrangement

Fa Dai[1], Ming Guo[1], Yina Shao[1], Chenghua Li[1,2]*

**1** State Key Laboratory for Managing Biotic and Chemical Threats to the Quality and Safety of Agro-products, Ningbo University, Ningbo, PR China, **2** Laboratory for Marine Fisheries Science and Food Production Processes, Qingdao National Laboratory for Marine Science and Technology, Qingdao, PR China

* lichenghua@nbu.edu.cn

**Data Availability Statement:** All relevant data are within the paper and its Supporting Information files.

## Abstract

We previously demonstrated that the flagellin of intracellular *Vibrio splendidus* AJ01 could be specifically identified by tropomodulin (Tmod) and further mediate p53-dependent coelomocyte apoptosis in the sea cucumber *Apostichopus japonicus*. In higher animals, Tmod serves as a regulator in stabilizing the actin cytoskeleton. However, the mechanism on how AJ01 breaks the AjTmod-stabilized cytoskeleton for internalization remains unclear. Here, we identified a novel AJ01 Type III secretion system (T3SS) effector of leucine-rich repeat-containing serine/threonine-protein kinase (STPKLRR) with five LRR domains and a serine/threonine kinase (STYKc) domain, which could specifically interact with tropomodulin domain of AjTmod. Furthermore, we found that STPKLRR directly phosphorylated AjTmod at serine 52 (S52) to reduce the binding stability between AjTmod and actin. After AjTmod dissociated from actin, the F-actin/G-actin ratio decreased to induce cytoskeletal rearrangement, which in turn promoted the internalization of AJ01. The STPKLRR knocked out strain could not phosphorylated AjTmod and displayed lower internalization capacity and pathogenic effect compared to AJ01. Overall, we demonstrated for the first time that the T3SS effector STPKLRR with kinase activity was a novel virulence factor in *Vibrio* and mediated self-internalization by targeting host AjTmod phosphorylation dependent cytoskeleton rearrangement, which provided a candidate target to control AJ01 infection in practice.

## Author summary

*Vibrio splendidus* AJ01 is the major pathogen for skin ulcer syndrome (SUS) in *Apostichopus japonicus*, nevertheless, its pathogenic mechanism remains unknown. Eukaryotic-like factors play a crucial role in bacterial virulence by targeting hosts. Despite their significance, the eukaryotic factors in *Vibrio splendidus* have not been studied, and the

**Funding:** This work was supported by the National Natural Science Foundation of China (32073003), the Fujian Aquatic Seed Industry Innovation and Industrialization Project (2021FJSC2Y03), and K. C. Wong Magna Fund in Ningbo University to CL. The funders had no role in study design, data collection and analysis, decision to publish, or preparation of the manuscript.

mechanism by which they interact with the host remains unclear. In this study, we found an eukaryotic-like factor STPKLRR of *V. splendidus* for the first time. Furthermore, STPKLRR, selected by T3SS, can phosphorylate Tmod at S52, dissociate the Tmod/actin complex, cause cytoskeleton rearrangement and promote the internalization of *V. splendidus*. Our findings provide insight into the mechanisms underlying the internalization of *V. splendidus* and advance our knowledge of the general biology of pathogen-host interactions.

## Introduction

Obligate and facultative intracellular bacteria have developed numerous strategies to alter the cell membrane to promote self-internalization, then achieve intracellular survival and replication [1]. Many pathogenic bacteria deliver bacterial virulence proteins into the host cell; this approach is an efficient method for hijacking the cell membrane and subverting the endocytic trafficking pathway to infect the host successfully [2, 3]. In general, the cellular processes that mediate bacteria internalization are driven by cytoskeletal networks [4]. The actin cytoskeleton, which is involved in numerous cellular motile events, such as cell membrane remodeling, endocytic pathway trafficking, and phagocytosis, is the potential target of various pathogen effectors [5]. Some effectors have been previously demonstrated to modify the actin cytoskeleton directly. Photox secreted by the *Photorhabdus luminescens* T6SS system directly targets all actin isoforms to inhibit the regular polymerization of actin filaments [6]. A high dose of the ACD domain of the *Vibrio cholerae* toxin MARTX can depolymerize actin into dimers, trimers, and macromolecular oligomers in target cells *in vitro* and is highly effective in inhibiting the dynamics of tandem actin-binding proteins [7]. Numerous bacterial effectors target and modify actin. They include SpvB from *Salmonella enterica* [8], the iota and C2 toxins from *Clostridium perfringens* [9], and VgrG1 from *V. cholerae*. Other effectors usually modify the actin cytoskeleton by altering actin-related proteins, mainly Rho-GTPases [10]. *Clostridium difficile* toxin A causes the disruption of the actin cytoskeleton through Rho-GTPase monoglucosylation, which induces the transcriptional upregulation of the p38-MAPK pathway [11]. The *Salmonella* effectors SopE, SopE2, and SopB trigger a burst of actin polymerization by activating the Rho-GTPases of the host cell in a redundant manner; this effect, in turn, can cause a defense response by activating the MAPK and NF-κB signaling cascade [12]. IcsA from *Listeria monocytogenes* can directly bind to neural Wiskott-Aldrich syndrome proteins to regulate actin polymerization and activate the actin-related protein 2/3 complex to regulate microfilament assembly [13]. Tmod, the only pointed end-capping protein of F-actin, constitutes and stabilizes the actin cytoskeleton and is also an ideal target for pathogen internalization [14]. However, to our knowledge, pathogen effectors targeting Tmod have not been identified, and their corresponding mechanism remains unclear.

*Vibrio splendidus* AJ01 is a facultative intracellular bacterium that is a common pathogen of mariculture species, particularly *Apostichopus japonicus* [15]. Type III secretion system (T3SS), which is closely related to effector factor secretion, pathogenicity, and outer membrane vesicles (OMVs), have been identified in this pathogen [16, 17]. Recently, we reveal that the flagellin of AJ01 activates the AjTmod-p38-MAPK pathway and induces coelomocyte apoptosis in *A. japonicus* [18]. In higher animals, Tmod serves as a regulator in stabilizing the actin cytoskeleton [19]. However, the mechanism via which AJ01 breaks through the AjTmod-stabilized cytoskeleton for internalization remains unclear. In this study, we identified a novel *A. japonicus* Tmod-interacting protein of leucine-rich repeat-containing serine/threonine-protein

kinase (STPKLRR) from AJ01. This protein could be secreted by T3SS and translocated into the cytosol of coelomocytes during AJ01 infection. Furthermore, STPKLRR could phosphorylate AjTmod at serine 52 to promote the dissociation of AjTmod from actin and decrease the F-actin/G-actin ratio. This effect induced cytoskeletal rearrangement and further mediated AJ01 internalization. The results of this study collectively advanced our understanding of the action of AJ01 STPKLRR as a virulence effector in the modification of the host endogenous protein AjTmod and manipulate AJ01 internalization in an actin-cytoskeleton-dependent manner.

## Results

### AJ01 STPKLRR is a novel AjTmod-binding protein

Tmod has been shown to inhibit actin cytoskeleton depolymerization, which regulates the cytoskeleton in a developmental stage- and tissue-specific manner [20]. For the identification of effectors targeting AjTmod, we performed a pull-down assay by using recombinant GST-fused AjTmod (GSTTmod, the complete AjTmod sequence) as the target to identify the potential interacting protein from AJ01 total protein. An obvious differential band was detected (red framed) through sodium dodecyl sulfate polyacrylamide gel electrophoresis (SDS-PAGE) and further characterized by using mass spectrometry (S1A Fig). The top 20 potential proteins with the highest scores were listed (S1B Fig) and all proteins were listed in S1 Data. In our previous work, FliC was confirmed to bind to AjTmod and further promote apoptosis via the p38-MAPK pathway [18]. A potential protein of STPKLRR (GenBank no. MBY7732200.1) with the highest score was selected for further functional validation. The predicted STPKLRR protein possessed five LRR domains located from 35–168 aa and a serine/threonine kinase (STYKc) domain from 203–403 aa (Fig 1A). A reverse pull-down assay was performed by using GSTTmod, the GST-fused tropomodulin domain (GSTTro, 8–148 aa of AjTmod), the GST-fused LRR domain (GSTTLRR, 229–284 aa of AjTmod), and GST-coated beads with His-fused STPKLRR followed by analysis with GST- or His-labeled antibodies to confirm the interaction region between AjTmod and STPKLRR. The results indicated that GSTTmod and GSTTro proteins, but not GST or GSTTLRR proteins, could specifically bind to soluble His-fused STPKLRR (Fig 1B). This finding suggested that the tropomodulin domain is responsible for the interaction between STPKLRR and AjTmod.

### STPKLRR is a novel virulence effector of AJ01

A *STPKLRR* knockout strain (ΔSTPKLRR) was constructed through markerless in-frame deletion (S2A Fig) to address the function of STPKLRR in AJ01 infection. The control strain ΔSTPKLRR::pMP2444 and the complemented strain ΔSTPKLRR::pMP-STPKLRR were also constructed on the basis of the pMP2444 plasmid and served as the control (S2B Fig). S3A Fig illustrates that *STPKLRR* deletion had no effect on bacterial growth. Weight loss is considered as a means of host defense against pathogens [21]. Subsequently, all four strains were used for a 5-day infection experiment to determine the change of body weight. We found that the weight of sea cucumbers in the AJ01 and ΔSTPKLRR::pMP-STPKLRR groups significantly decreased from 150 ± 14 g to 55 ± 10 and 65 ± 6 g, respectively. The weights of the sea cucumbers in the ΔSTPKLRR and ΔSTPKLRR::pMP2444 groups decreased to 100 ± 7 and 115 ± 5 g, respectively (Fig 2A). Moreover, sea cucumbers infected with the ΔSTPKLRR and ΔSTPKLRR::pMP2444 strains exhibited a substantial delay in the time of death and an obvious decrease in mortality compared with those infected with the AJ01 and ΔSTPKLRR::pMP-STPKLRR strains. The highest mortality rate was shown in the AJ01 group, which had a total of 25 deaths and a survival rate of only 16.7%. ΔSTPKLRR::pMP-STPKLRR had the next-

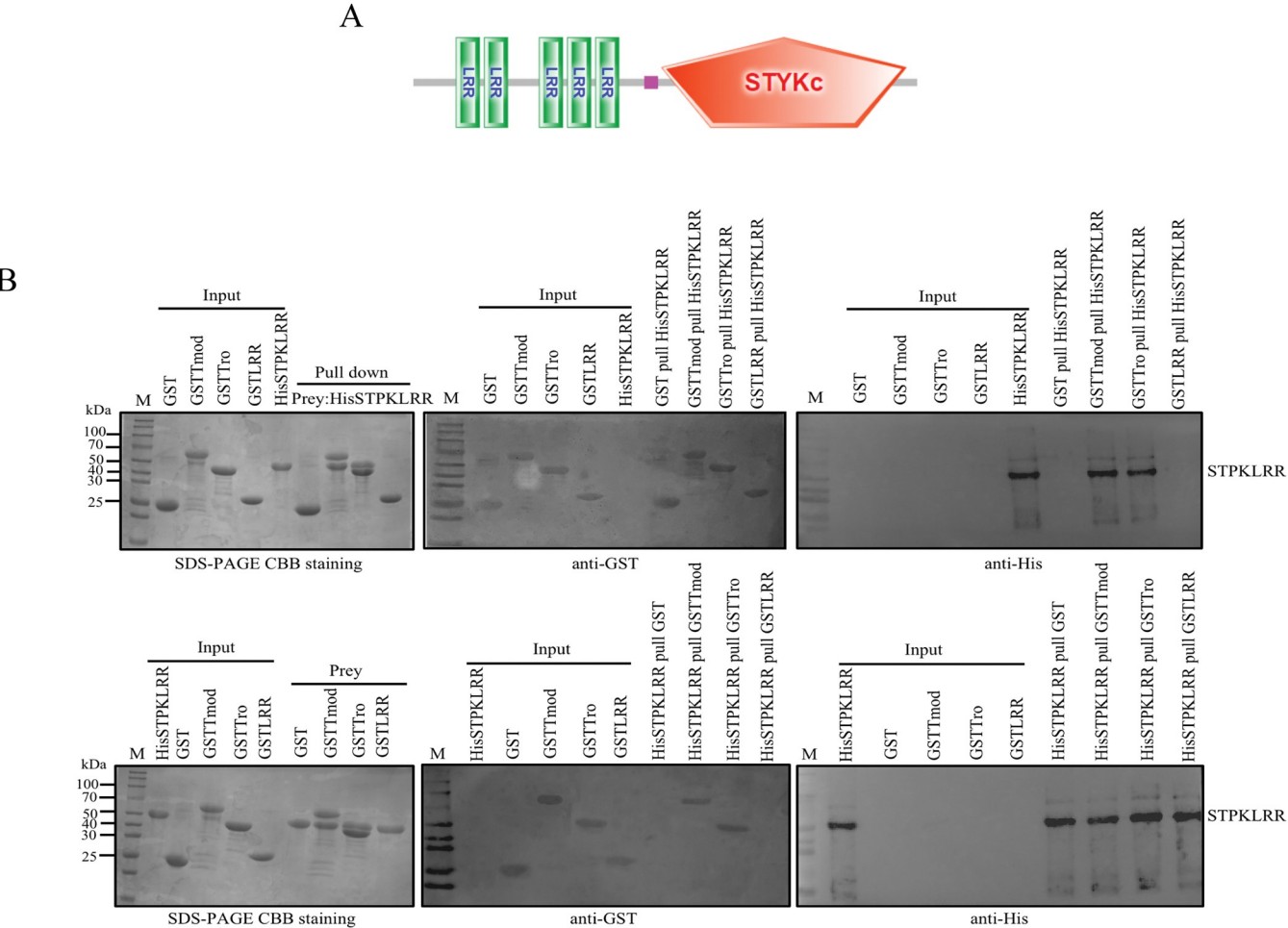

**Fig 1. STPKLRR is a novel Tmod-interacting protein in AJ01. (A)** The domain structure of STPKLRR with the highest score was analyzed by SMART. **(B)** For the further validation of the interaction between AjTmod and STPKLRR, different regions of AjTmod were fused with GST-tag, (GSTTro [GST-fused tropomodulin domain], GSTTLRR [GST-fused LRR domain]), and their interaction with His-fused STPKLRR was analyzed through pull-down assays. GSTTmod, GST-fused full-length AjTmod; GSTTro, GST-fused Tropomodulin domain of AjTmod; GSTLRR, GST-fused LRR domains of AjTmod. Upper panel: M: Protein marker; Lane 1, purified GST; Lane 2, purified GSTTmod; Lane 3, purified GSTTro; Line 4, purified GSTLRR; Line 5, purified STPKLRR; Lane 6, GST pulled STPKLRR elution sample; Lane 7, GSTTmod pulled STPKLRR elution sample; Lane 8, GSTTro pulled STPKLRR elution sample; Lane 9, GSTLRR pulled STPKLRR elution sample. Lower panel: Lane 1, purified STPKLRR; Line 2, purified GST; Lane 3, purified GSTTmod; Lane 4, purified GSTTro; Line 5, purified GSTLRR; Line 6, STPKLRR pulled GST elution sample; Line 7, STPKLRR pulled GSTTmod elution sample; Line 8, STPKLRR pulled GSTTro elution sample; Line 9, STPKLRR pulled GSTLRR elution sample. The second panel in each row presents western blotting analysis of the GST tag signal performed using a GST-labeled mouse monoclonal antibody. The third panel in each row presents western blotting analysis of the His tag signal performed using a His-labeled mouse monoclonal antibody.

highest mortality rate with a total of 22 deaths and the survival rate of 26.7%. The ΔSTPKLRR and ΔSTPKLRR::pMP2444 groups had the lowest individual mortalities of 16 and 13, respectively, and the survival rates of 46.7% and 53.3%, respectively (Fig 2B).

Six tissues of body walls, tentacles, muscles, intestines, and respiratory trees were subjected to histological observation at 72 h after pathogen infection to further confirm the pathogenic effect of STPKLRR. The results revealed that in all examined tissues from the AJ01 and ΔSTPKLRR::pMP-STPKLRR groups, the integrity of the tissue structure was destroyed and tissue connections had loosened (Figs 2C and S3B). The most obvious pathological change was detected in the tissues of the muscle and intestine, which presented dense muscle fibers with holes and damaged intestinal mucosa (Fig 2C, black arrows). However, sea cucumbers from

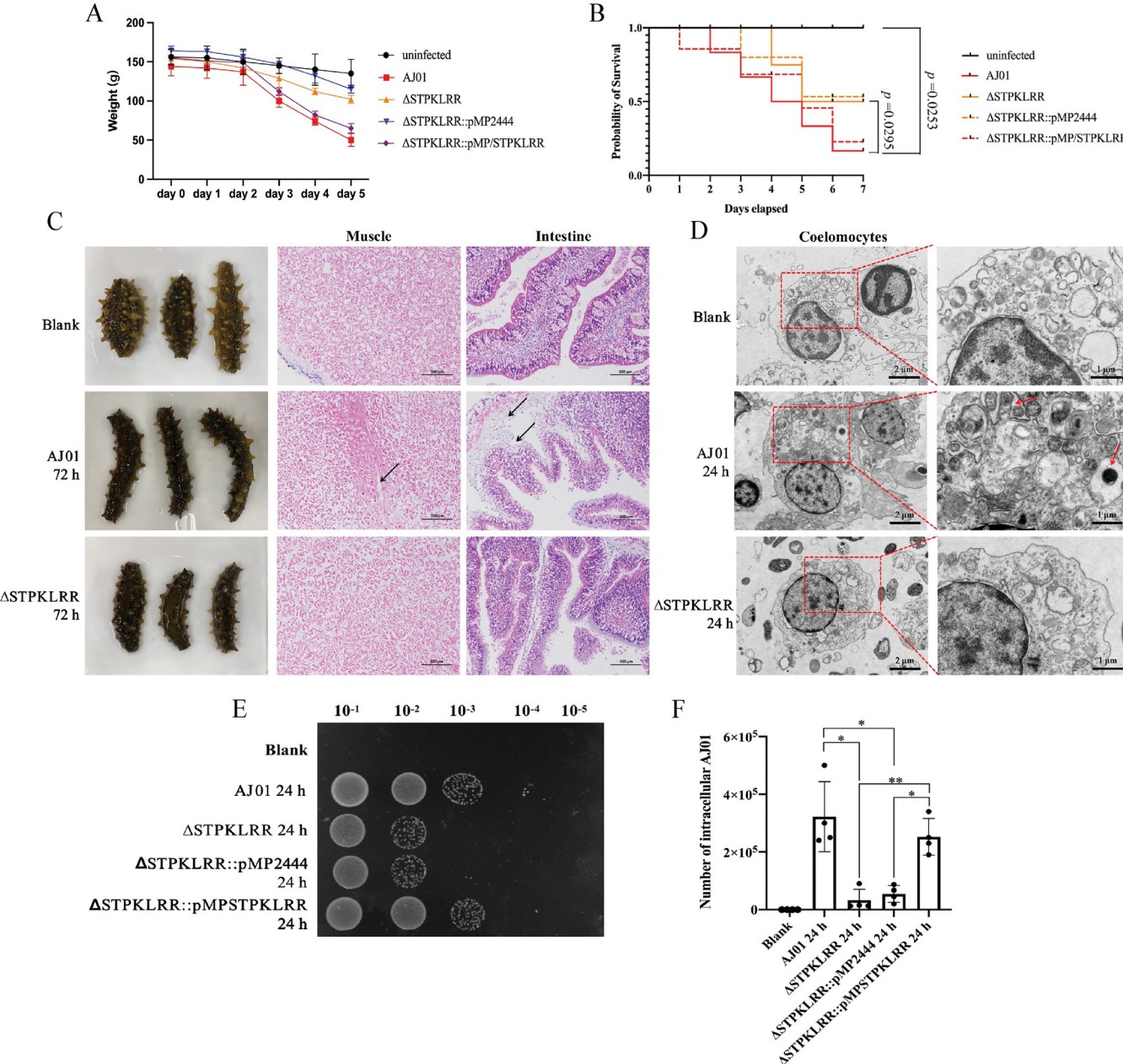

**Fig 2. STPKLRR is an essential virulence factor for AJ01 infection.** A STPKLRR knockout strain (ΔSTPKLRR) was constructed via markerless in-frame deletion. The control strain ΔSTPKLRR::pMP2444 and the complemented strain ΔSTPKLRR::pMP-STPKLRR served as the control. **(A)** After 5 days of infection with the four strains, significant reductions in body weights were detected in sea cucumbers infected with the AJ01 and ΔSTPKLRR::pMP/STPKLRR strains. **(B)** A substantial delay in the time of death and an obvious decrease in mortality was found in the ΔSTPKLRR and ΔSTPKLRR::pMP2444 strains. **(C)** Muscles, intestines were subjected to histological observation at 72 h after AJ01 and ΔSTPKLRR infection to further confirm the pathogenic effect. Black arrows represent areas of tissue damage. Scale bar, 100 μm. **(D)** Intracellular AJ01 and ΔSTPKLRR (red arrows) was detected by transmission electron microscopy, and the intracellular AJ01 in the coelomocytes was quantified through serial dilution **(E)** after infection with the four strains for 24 h. **(F)** The graph is representative of three independent assays, $^*p < 0.05$, $^{**}p < 0.01$. The isolated intracellular bacteria were subjected to 16S rDNA sequencing to confirm it was AJ01.

the ΔSTPKLRR::pMP2444 and ΔSTPKLRR groups exhibited no obvious tissue morphological changes. Transmission electron microscopy more frequently identified the intracellular AJ01-like structure in the AJ01 and ΔSTPKLRR::pMP-STPKLRR groups than in the ΔSTPKLRR and ΔSTPKLRR::pMP2444 groups (Figs 2D and S3C, red arrows). For the further

confirmation of these AJ01-like structures and the changes in the number of intracellular bacteria, the collected intracellular bacteria were cultured through serial dilutions (Fig 2E) and analyzed via 16S rDNA sequencing, which found sequences identical to the AJ01 sequence. Clonal counting revealed that the numbers of intracellular AJ01 in groups AJ01 and ΔSTPKLRR::pMP-STPKLRR reached $3 \times 10^5$ and $2.1 \times 10^5$ CFU/mL, respectively, whereas that in the ΔSTPKLRR and ΔSTPKLRR::pMP2444 groups were only $2 \times 10^4$ and $2.7 \times 10^4$ CFU/mL, respectively (Fig 2F).

## STPKLRR is secreted via T3SS

Pathogen-secreted effectors directly target the host to facilitate host interaction [22], and many bacteria have evolved complex multiprotein machines to deliver effector proteins to target cells to achieve infection [23, 24]. The total and secreted proteins of AJ01 were extracted and subjected to SDS-PAGE to investigate whether STPKLRR is a secreted protein. STPKLRR was identified from the secreted proteins of AJ01 by using Western blot analysis with anti-STPKLRR. Chaperone protein DnaK, which was used as a marker of AJ01 intracellular proteins, was not detected under the same conditions (Fig 3A). The β-lactamase reporter system assay revealed that in AJ01-infected coelomocytes, STPKLRR could be translocated into the host cells in a time-dependent manner (Fig 3B). Moreover, we confirmed that the mRNA expression level of STPKLRR increased by 22.6%, 36.3%, and 60.5% after 1, 3, and 6 h of incubation with coelom fluid compared to the untreated group, respectively (S4A Fig). Under the same condition, the protein expression and secretion of STPKLRR similarly increased (S4B Fig). Similarly, immunofluorescence microscopy revealed that the fluorescence signal of STPKLRR appeared in coelomocytes at 1 h and gradually increased at 3 h with an MOI of 100. No fluorescence signal was detected in the ΔSTPKLRR strain infection group under the same condition (Fig 3C). Importantly, fluorescence signals between STPKLRR and AjTmod were found to colocalize in coelomocytes (Fig 3C). Consistently, the number of AJ01 in coelomocytes was increased accompanied with the increase of intracellular STPKLRR (S4C Fig). Furthermore, Western blot analysis mainly detected STPKLRR in coelomocyte cytosolic proteins in the AJ01 and ΔSTPKLRR::pMP-STPKLRR groups but not in the ΔSTPKLRR::pMP2444 and ΔSTPKLRR groups (Fig 3D).

Two major secretion pathways of T3SS and OMVs in AJ01 were investigated to further determine the secretory pathway of STPKLRR. Six T3SS inhibitors, namely, cinnamaldehyde, salicylidene acylhydrazide, phenoxyacetamide, piericidin A, quercetin, and quinine [25], were selected and their optimal concentrations were determined (S5B–S5F Fig). The secretion and the antibody specificity (S5A Fig) of Hop, a marker of T3SS secretion in AJ01, was also determined. A significant reduction in Hop secretion was detected under treatment with the optimal concentrations of cinnamaldehyde, salicylidene acylhydrazide, and phenoxyacetamide, but not under treatment with the optimal concentrations of piericidin A, quercetin, and quinine. Similarly, the secretion of STPKLRR was also significantly inhibited under the same conditions (Figs 3E and S5G–S5K). The β-lactamase reporter system assay consistently showed that the translocation of STPKLRR into coelomocytes significantly decreased after treatment with 50 μM cinnamaldehyde (Fig 3F). All these results supported that the secretion of STPKLRR from AJ01 was dependent on T3SS.

Bacterial vesicles have also been confirmed to be involved in effector secretion [26]. Therefore, the secretion of STPKLRR based on bacterial vesicles was also examined. The OMVs of AJ01 were fractionated through density gradient ultracentrifugation as described by Park et al [27]. The AJ01 supernatant and pellet fractions were used as controls. Fig 3G showed that STPKLRR was detected in the supernatant and pellet, but not in vesicles. Outer membrane

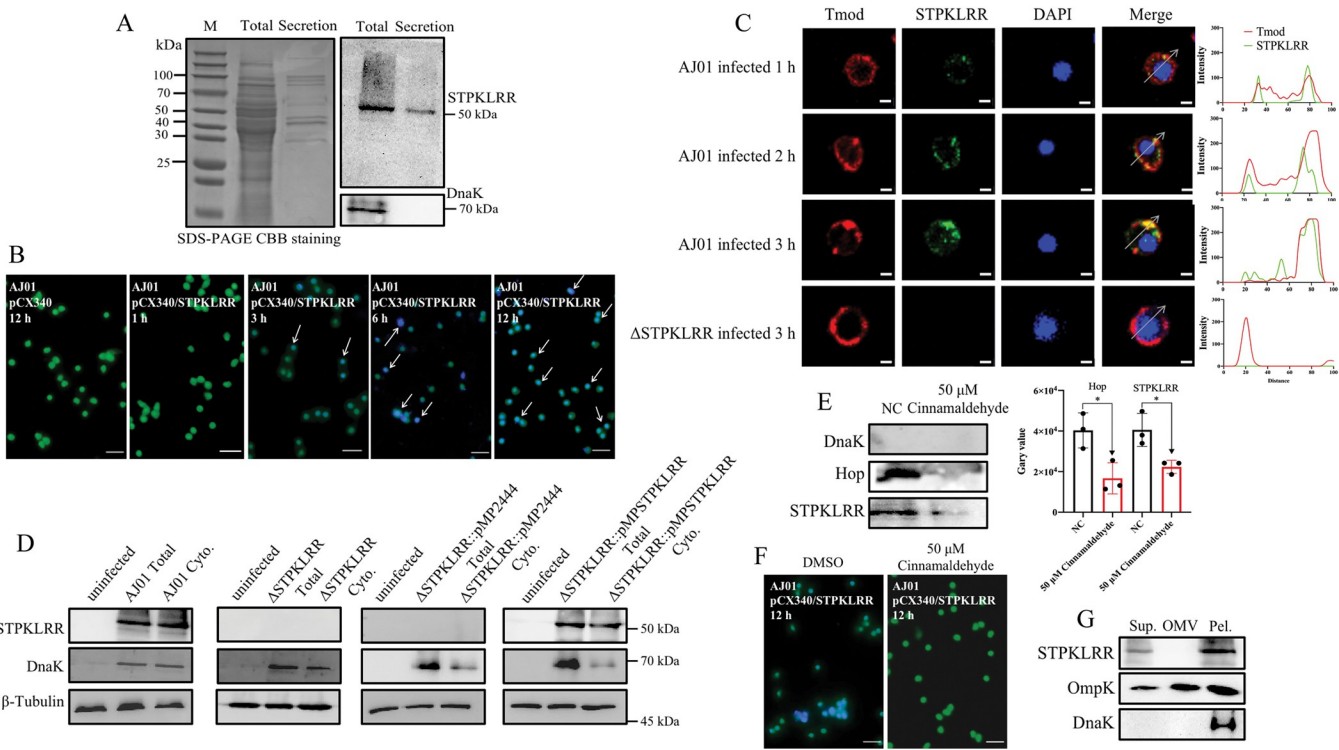

**Fig 3. STPKLRR is an effector secreted by AJ01 T3SS. (A)** The total protein and secreted protein of AJ01 were extracted and separated by SDS-PAGE. Western blot was uesd to detect STPKLRR in total and secreted proteins with anti-STPKLRR. DnaK was used as a marker of intracellular proteins**. (B)** β-Lactamase assay was used to determine whether STPKLRR was secreted in coelomocytes after 1, 3, 6, and 12 h of AJ01 infection. pCX340 vector without any insertion served as the control. After infection, coelomocytes were loaded with CCF4-AM. Translocation of STPKLRR into the coelomocytes results in the cleavage of CCF4-AM, causing the emission of blue fluorescence. Uncleaved CCF4-AM emits green fluorescence. Scale bar, 50 μm. **(C)** To further address the spatial distribution of STPKLRR, coelomocytes were infected with AJ01 for 1, 2, and 3 h or ΔSTPKLRR for 3 h at a MOI of 100. The STPKLRR signal was significantly enhanced in the cytoplasm after AJ01 infection. No STPKLRR signal was detected in the ΔSTPKLRR infection group. Scale bar, 3 μm. **(D)** Intracellular translocation of STPKLRR was determined through Western blot analysis after AJ01 or ΔSTPKLRR infection for 3 h at a MOI of 100. DnaK served as a bacterial cytosolic marker, and β-Tubulin was used as a marker of cytosolic proteins. **(E)** The optimal concentration of the T3SS inhibitor cinnamaldehyde was determined. The secretion of Hop (a marker of *Vibrio* T3SS secreted proteins) and STPKLRR was further analyzed through Western blot analysis in the same conditions. Band density was quantified by using ImageJ. Data (means ± SD) are representative of at least 3 experiments. Asterisks indicate significant differences (*$p < 0.05$). **(F)** The secretion of STPKLRR into coelomocytes after treatment with 50 μM cinnamaldehyde was further validated by performing the β-lactamase assay. **(G)** To investigate whether STPKLRR is a secreted from vesicles, proteins from AJ01 supernatants (Sup.), outer membrane vesicles (OMVs), and pellets (Pel.) were analyzed through Western blot analysis. OmpK was used to probe of vesicles outer membrane protein.

protein K (OmpK), a marker of vesicles, could be identified in OMVs, and DnaK, a marker of intracellular proteins, was found in pellets.

## Intracellular STPKLRR directly phosphorylates AjTmod at S52

Phosphorylation modification by protein kinases is a common way to induce subsequent signal transduction [28]. The change in AjTmod phosphorylation levels was investigated in AJ01- or ΔSTPKLRR infected sea cucumbers to determine the phosphorylation function of STPKLRR in coelomocytes. Phos-tag SDS-PAGE was utilized to identify the number of modified phosphorylation sites [29]. We found that AjTmod in the AJ01-infected group was significantly phosphorylated relative to that in the uninfected group (0 h). Compared with 0 h and 3 h, the two sites of AjTmod were significantly phosphorylated at 12 h and 24 h. In the ΔSTPKLR infection group, only one site of AjTmod was phosphorylated at 12 and 24 h (Fig 4A, red arrows). All these results supported that one site of AjTmod phosphorylation was mediated by STPKLRR during AJ01 infection. The phosphorylation levels of AjTmod were

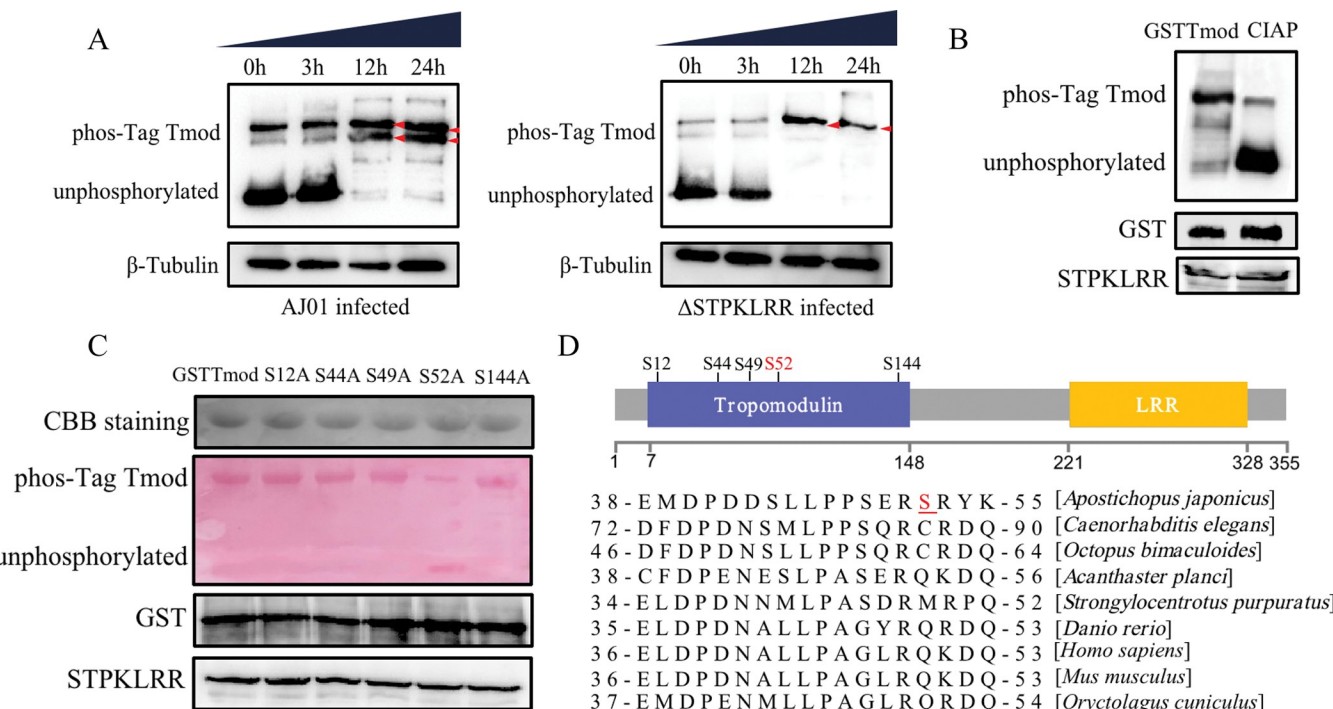

**Fig 4. Intracellular STPKLRR directly phosphorylates AjTmod at S52. (A)** To determine the AjTmod phosphorylation function by STPKLRR, Phos-tag SDS-PAGE was used to investigate the changes in AjTmod phosphorylation levels in coelomocytes after AJ01 or ΔSTPKLRR infection for 3, 12, and 24 h. Phosphorylated AjTmod were indicated by red arrows and non-phosphorylated AjTmod served as the control. **(B)** The phosphorylation levels of AjTmod were assayed *in vitro* with the phosphorylation inhibitor of CIAP to confirm that STPKLRR directly phosphorylated AjTmod. **(C)** Five serine sites in the tropomodulin domain of AjTmod were mutated into alanine, namely, S12A, S44A, S49A, S52A, and S144A, to confirm the potential phosphorylation site. In the *in vitro* kinase phosphorylation assay, only the S52A mutant was not phosphorylated by STPKLRR. **(D)** The evolutionary conservation of AjTmod S52 was further analyzed with Tmods from *Caenorhabditis elegans* (NP_491735.1), *Octopus bimaculoides* (XP_014786736.1), *Acanthaster planci* (XP_022108191.1), *Strongylocentrotus purpuratus* (XP_030829490.1), *Danio rerio* (XP_001920602.1), *Homo sapiens* (NP_001159588.1), *Mus musculus* (EDL02383.1), and *Oryctolagus cuniculus* (XP_008267220.1).

assayed *in vitro* with the phosphorylation inhibitor of calf intestine alkaline phosphatase (CIAP) to confirm that STPKLRR directly phosphorylated AjTmod. The results indicated that CIAP supply could depress STPKLRR-mediated AjTmod phosphorylation (Fig 4B). The serine residues are the major target for protein phosphorylation and are also critical to signal transduction [30]. Given that STPKLRR targets the tropomodulin domain of AjTmod, we selected five serine sites in the tropomodulin domain for mutation into alanine, namely, S12A, S44A, S49A, S52A, and S144A. We found that GSTTmod and the four mutants S12A, S44A, S49A, and S144A could be phosphorylated by recombinant STPKLRR *in vitro*. However, the S52A mutant lost the ability for phosphorylation by STPKLRR (Fig 4C). Surprisingly, the phosphorylation site was unique to AjTmod and was not found in several other invertebrates and higher mammals. (Fig 4D).

To confirm STPKLRR phosphorylating AjTmod rather than intracellular AJ01, we treated coelomocytes with recombinant STPKLRR or His tag protein for 12 h. As shown in S6A Fig, only one phosphorylated site of AjTmod was detected in the STPKLRR treated group. Furthermore, the FITC-labeled fluorescent microspheres (2 μm in diameter, Sigma) was used to assay the coelomocyte phagocytosis activity under the same condition. The results showed that phagocytosis rate in STPKLRR treated group was significantly induced compared to the His tag treated group (S6B and S6C Fig). These results supported that phosphorylation of AjTmod was modified by STPKLRR rather than intracellular AJ01.

## Phosphorylated AjTmod is dissociated from actin and locates in cytoplasma

In metazoans under normal conditions, Tmod could specifically bind to F-actin to stabilize the cytoskeleton [31, 32]. The cellular location of AjTmod was investigated under different conditions to further assess the effect of phosphorylated AjTmod on its interaction with actin. Immunofluorescence microscopy revealed that in the early stages of AJ01 infection (MOI = 100), STPKLRR could colocalize with AjTmod and actin (Fig 5A). With the prolongation of infection time, AjTmod gradually transferred from the cell membrane to the cytoplasm (Fig 5B). DiIC18 (3) (Dil), a lipophilic membrane dye, was used to label the cell membrane (Fig 5B). In the ΔSTPKLRR group, the cellular location of AjTmod did not show any significant changes throughout the whole stage of infection (Fig 5B, bottom panel). Coelomocyte subcellular fractionation analysis revealed that in the AJ01 infection group, the fraction of AjTmod in the cell membrane decreased from 67.7% to 15%, whereas that of AjTmod in the cytoplasm increased from 32.3% to 85% (Fig 5C and 5D). However, sodium/potassium-transporting ATPase subunit alpha-1 (ATPA1), a cell membrane protein marker, was mainly

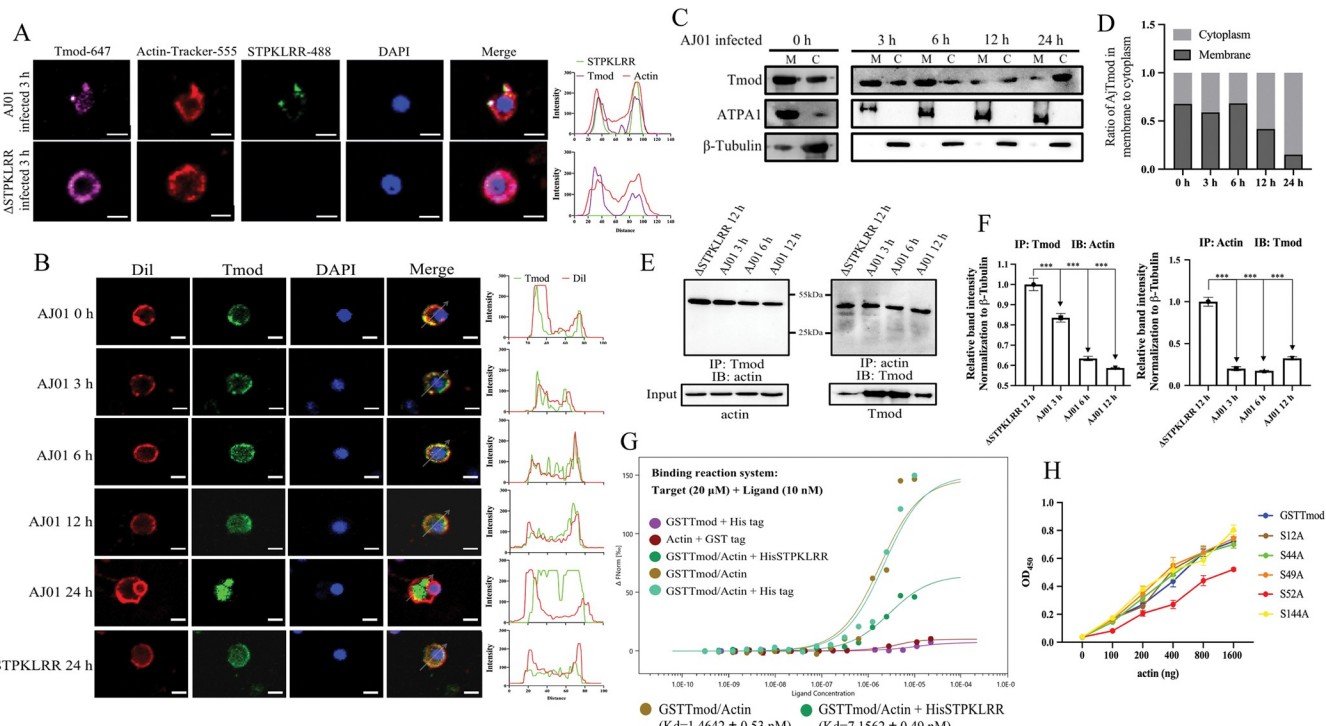

**Fig 5. Phosphorylated AjTmod is dissociated from actin and locates in cytoplasma.** (A) The coelomocyte locations of AjTmod, STPKLRR, and actin were investigated after STPKLRR or AJ01 infection with MOI = 100 to further assess the effect of phosphorylated AjTmod on its interaction with actin. Scale bar, 5 μm. (B) The spatial location of AjTmod was investigated through laser confocal technology after AJ01 infection for 3, 6, 12, and 24 h. The membrane was labeled with DiIC18(3) (Dil), a lipophilic membrane dye. The group infected with ΔSTPKLRR for 24 h served as the control group. Scale bar, 5 μm. (C) The coelomocyte membrane (M) and cytoplasmic (C) fractions were isolated from above condition, and the intracellular translocation of AjTmod was determined through Western blot analysis. Sodium/potassium-transporting ATPase subunit alpha-1 (ATPA1) was used as a membrane protein marker, and β-Tubulin was used as a cytoplasmic protein marker. (D) The ratio of AjTmod in the cell membrane and cytoplasm was shown. (E) To further validate the interaction between AjTmod and actin, coimmunoprecipitation was performed with AjTmod- and actin-labeled antibodies. (F) The band densities of AjTmod and actin were quantified and normalized to those of actin and AjTmod. Data (means ± SD) are representative of at least three experiments. Asterisks indicate significant differences (***$p < 0.001$). (G) The dissociation constant ($K_d$) of GST-fused AjTmod and His-fused actin with or without His-fused STPKLRR was determined by using MST to quantify the effect of STPKLRR on the stability of the AjTmod-actin complex. The $K_d$ of the GST-fused Tmod and His-fused actin was 1.4642 ± 0.53 nM. After the addition of STPKLRR, the $K_d$ changed to 7.1562 ± 0.49 nM. GST and His tags were used as the control group. (H) ELISA was applied to further determine the role of S52 in AjTmod-actin interaction.

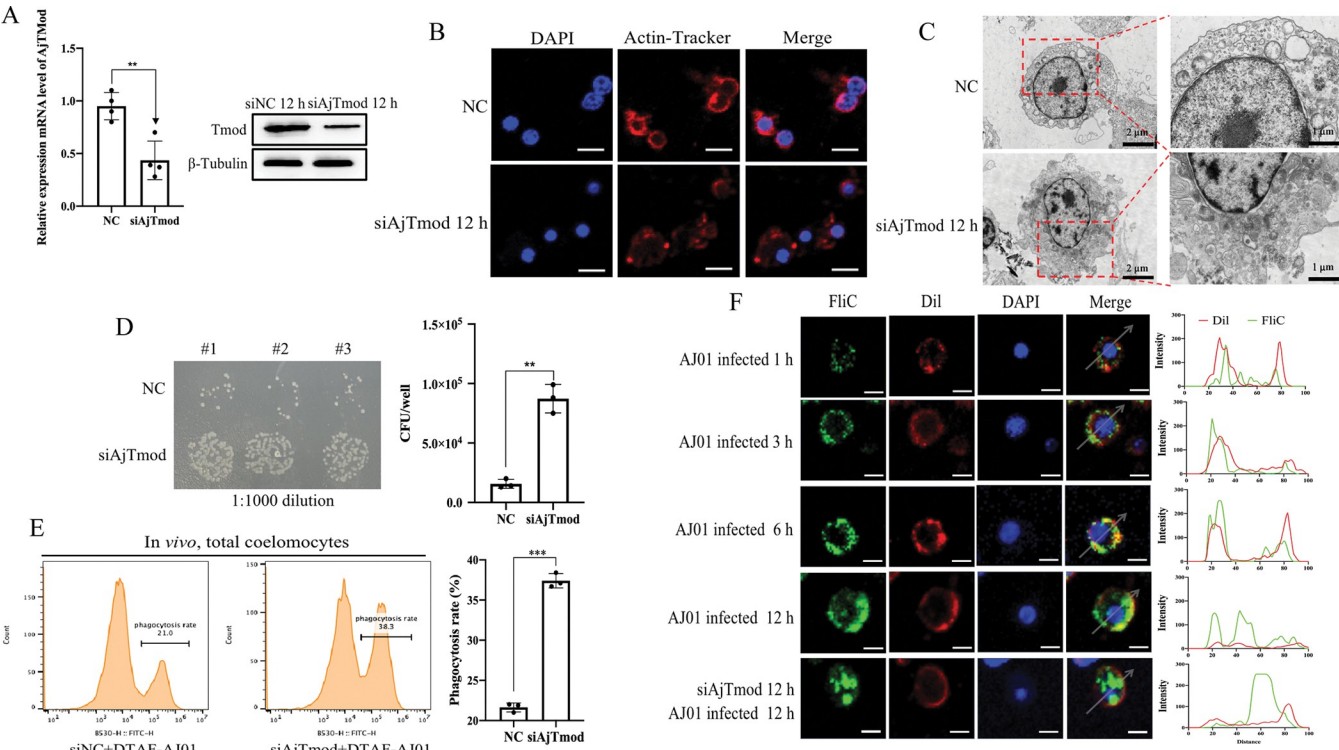

**Fig 6. AjTmod modulates coelomocytes phagocytosis. (A)** AjTmod silencing by specific siRNAs was performed *in vivo* and was confirmed by qRT-PCR and Western blot analyses to determine the function of AjTmod in phagosome formation. Actin cytoskeleton rearrangement under this condition was further analyzed by immunofluorescence **(B)**, and membrane ruffling was analyzed through transmission electron microscopy **(C)**. The count of intracellular AJ01 **(D)** and the phagocytosis rate **(E)** were determined to further determine the relationship between membrane rearrangement and phagocytosis. **(F)** With AJ01 infection at 1 h, 3 h, 6 h, and 12 h, the number of intracellular AJ01 gradually increased. After AjTmod silenced and 12 h AJ01 infection, the number of intracellular AJ01 was significantly more than that of 12 h AJ01 infection. FliC, the flagellin of AJ01, was used to lebel AJ01.

detected in the cell membrane fraction and displayed no significant change under AJ01 infection (Fig 5C). β-Tubulin, a marker of cytoplasmic protein, was also mostly detected in the cytoplasm fraction and showed no significant change in expression during AJ01 infection (Fig 5C). All these results supported the migration of AjTmod from the cell membrane to the cytoplasm in response to AJ01 infection.

We further used coimmunoprecipitation to verify the shift in AjTmod translocation in AJ01- and ΔSTPKLRR-challenged coelomocytes. The coimmunoprecipitation of AjTmod and actin showed that in the AJ01 group, the contents of protein bound to AjTmod or actin gradually decreased with the prolongation of infection time. However, the binding ability of AjTmod and actin did not change in the ΔSTPKLRR group (Fig 5E and 5F). The change in the dissociation constant ($K_d$) between GSTTmod and His-fused actin before and after STPKLRR administration was assessed by microscale thermophoresis assay (MST) to assess whether the dissociation of AjTmod and actin was caused by STPKLRR. The reaction system was Target (20 μM) + Ligand (10 nM). The results indicated that $K_d$ increased from 1.4642 ± 0.53 nM to 7.1562 ± 0.49 nM after the addition of STPKLRR (Fig 5G). Moreover, ELISA revealed that the binding ability of the S52A mutant to His-fused actin had significantly reduced relative to that of AjTmod and the other four mutants (Fig 5H).

To determine the effect of AjTmod dissociation from actin mediating actin cytoskeleton, immunofluorescence and transmission electron microscope analysis were performed. The results showed that damage of cell membrane morphology including depolymerization and

rearrangement appeared in the AJ01 infected group, not ΔSTPKLRR group (S7A and S7B Fig). These results indicated that the binding stability of AjTmod-actin played an important role in stabilizing the coelomocyte cytoskeleton.

## AjTmod modulates coelomocyte phagocytosis

Tmod plays essential roles in phagocytosis as an important cytoskeleton component that can inhibit actin depolymerization and maintain the cytoskeleton [33]. AjTmod-interacting proteins were identified through a pull-down assay and further characterized by mass spectrometry. All the AjTmod- interacting proteins were listed in S2 Data. The results indicated that AjTmod was closely related to phagosome formation given that phagosome-related proteins were enriched in GO (S8A Fig), KEGG (S8B Fig), and PPI (S8C Fig) analyses. Consistently, the mRNA level of AjTmod significantly decreased to 43.5% at 12 h in the groups with AjTmod silencing by specific siRNA *in vivo* relative to that in the control group and the protein level of AjTmod also decreased in the siAjTmod group (Fig 6A). Under this condition, the actin cytoskeleton showed obvious depolymerization and rearrangement, and the cell

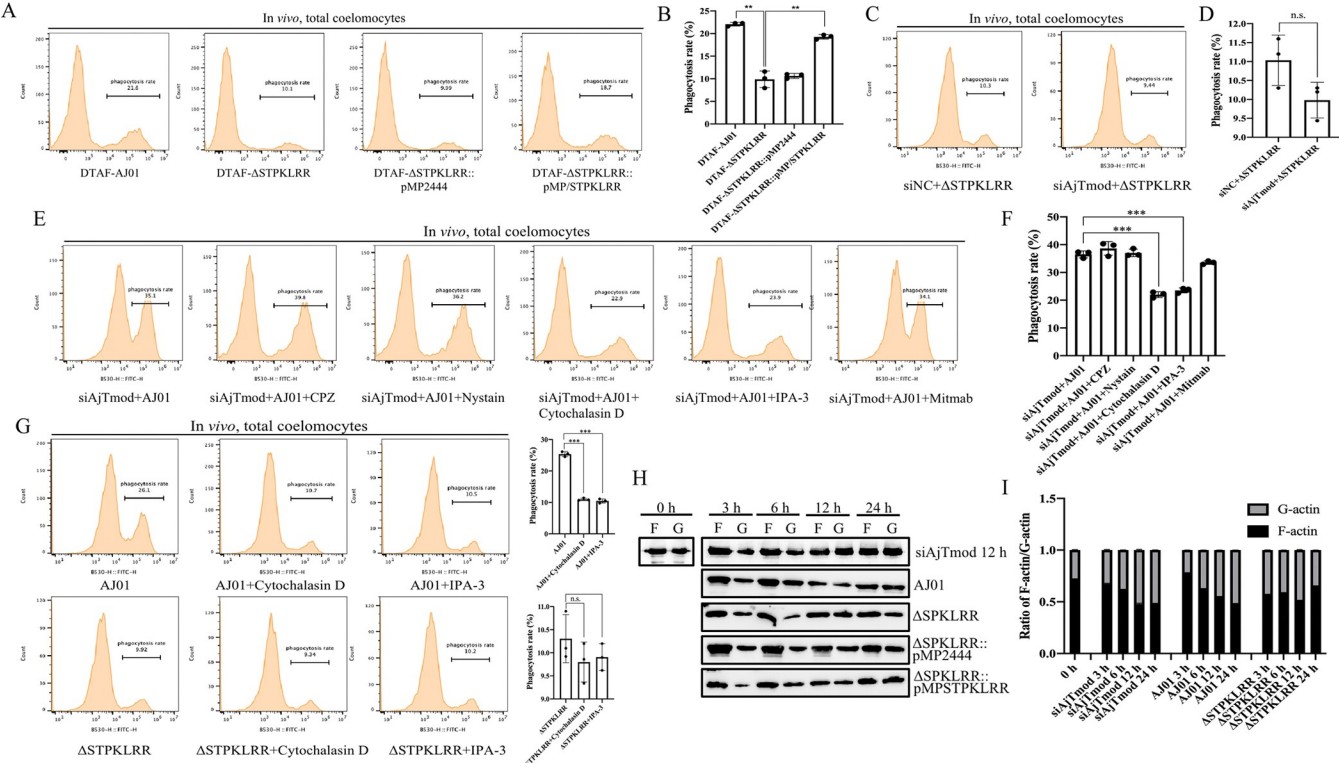

**Fig 7. STPKLRR promotes AJ01 internalization via macropinocytosis and actin-dependent pathways by targeting AjTmod. (A)** The phagocytosis of coelomocytes was determined after AJ01, ΔSTPKLRR, ΔSTPKLRR::pMP2444, and ΔSTPKLRR::pMP-STPKLRR infection to confirm whether STPKLRR could promote AJ01 internalization by targeting AjTmod. The corresponding values are shown in **(B)**. **(C)** Phagocytosis activity was determined after AjTmod knockdown and infection with the ΔSTPKLRR strain to determine whether AJ01 internalization is dependent on the interaction between AjTmod and STPKLRR. The corresponding values are shown in **(D)**. The experiment was performed with three independent assays, $**p < 0.01$. **(E)** Phagocytic activities were assayed through flow cytometry after AjTmod knockdown and treatment with the optimal concentration of five specific phagocytosis inhibitors to explore which endocytic pathway is required for STPKLRR-mediated AJ01 internalization. The corresponding values are presented in **(F)**. **(G)** Coelomocytes were treated with cytochalasin D and IPA-3, then challenged with the AJ01 or ΔSTPKLRR strains, respectively, to illustrate further that the STPKLRR-mediated internalization of AJ01 is mainly dependent on macropinocytosis and actin-dependent endocytic pathways. Phagocytic activities were determined and are shown in right panels. **(H)** The F-actin/G-actin ratio was determined through Western blot analysis after AjTmod knockdown for 12 h followed by infection with four AJ01 strains for 0, 3, 6, 12, and 24 h. The F-actin/G-actin ratio was calculated in **(I)**.

membrane also presented multiple protrusions and folds under immunofluorescence microscopy and transmission electron microscopy (Fig 6B and 6C).

Change in the actin cytoskeleton induces numerous cellular events, including intracellular vesicle trafficking and phagocytosis [6]. The intracellular bacterial colonization level in coelomocytes was investigated *in vitro* to test whether AjTmod was indeed related to the phagocytosis of AJ01. After 12 h of AjTmod knockdown followed by AJ01 infection for an additional 3 h, intracellular AJ01 was collected and spread on 2216E solid medium. The results showed that the numbers of intracellular bacteria in the siAjTmod group had significantly increased compared with those in the NC group (Fig 6D). Phagocytic activity *in vivo* was detected by flow cytometry under the above condition. Phagocytic activity in the siAjTmod group had significantly increased from 21.5% ± 0.5% to 37.5% ± 0.8% (Fig 6E). FliC immunofluorescence analysis revealed that intracellular AJ01 had significantly increased in coelomocytes after *AjTmod* knockdown (Fig 6F).

## STPKLRR promotes AJ01 internalization via the macropinocytosis and actin-dependent endocytic pathways by targeting AjTmod

All above results supported that STPKLRR promoted the disassociation of AjTmod from actin and AjTmod mediated cellular phagocytosis, which indicated that STPKLRR might promote AJ01 internalization by targeting AjTmod. We assayed coelomocyte phagocytosis after infection with the AJ01, ΔSTPKLRR, ΔSTPKLRR::pMP2444, and ΔSTPKLRR::pMP-STPKLRR strains to confirm the above hypothesis. We found that compared with AJ01, ΔSTPKLRR decreased the phagocytosis rate from 21.9% ± 0.3% to 9.7% ± 1.8%, whereas the complemented strain ΔSTPKLRR::pMP-STPKLRR reverted the phagocytosis rate to 19.2% ± 0.5% (Fig 7A and 7B). We examined phagocytosis after AjTmod knockdown and ΔSTPKLRR infection to further determine whether the internalization of AJ01 is dependent on the interaction between AjTmod and STPKLRR. We found that the phagocytosis rates in the siNC +ΔSTPKLRR and siAjTmod+ΔSTPKLRR groups showed no significant changes (Fig 7C and 7D). All these results indicated that STPKLRR was critical for AJ01 internalization by targeting AjTmod. What's more, the internalization of AJ01 led to an increase in lysosomal activity. Conversely, the loss of STPKLRR resulted in a decrease in lysosomal activation. These findings suggested a strong correlation between STPKLRR-mediated internalization and intracellular lysosomal activation (S9 Fig).

In our previous work, we found that AJ01 could enter coelomocytes through clathrin-, macropinocytosis-, actin-, and dynamin-dependent pathways [34]. We used five specific inhibitors to inhibit various endocytic pathways to explore which endocytic pathway was required for STPKLRR-mediated AJ01 internalization. Under AjTmod knockdown and treatment with five specific inhibitors, cytochalasin D and IPA-3 could block AjTmod-mediated phagocytic activity. The rate of AJ01 internalization in the cytochalasin D- and IPA-3-treated groups decreased from 36.2% ± 1.1% to 21.8% ± 1.0% and 23.3% ± 0.9%, respectively (Fig 7E and 7F), whereas the siAjTmod+AJ01+chlorpromazine (CPZ), siAjTmod+AJ01+nystain, and siAjTmod+AJ01+mitmab groups showed no significant changes (Fig 7E and 7F). Cytochalasin D and IPA-3 inhibit the macropinocytosis and actin-dependent endocytic pathways, respectively, which are closely related to the actin-mediated endocytic pathway [4]. We treated coelomocytes with cytochalasin D and IPA-3, then challenged them with the AJ01 and ΔSTPKLRR strains to verify that the STPKLRR-mediated internalization of AJ01 mainly occurs through macropinocytosis and actin-dependent endocytic pathways. We found that the phagocytosis rate of AJ01 decreased from 25.3% ± 0.8% to 11.1% ± 0.4% and 10.4% ± 0.7% after cytochalasin

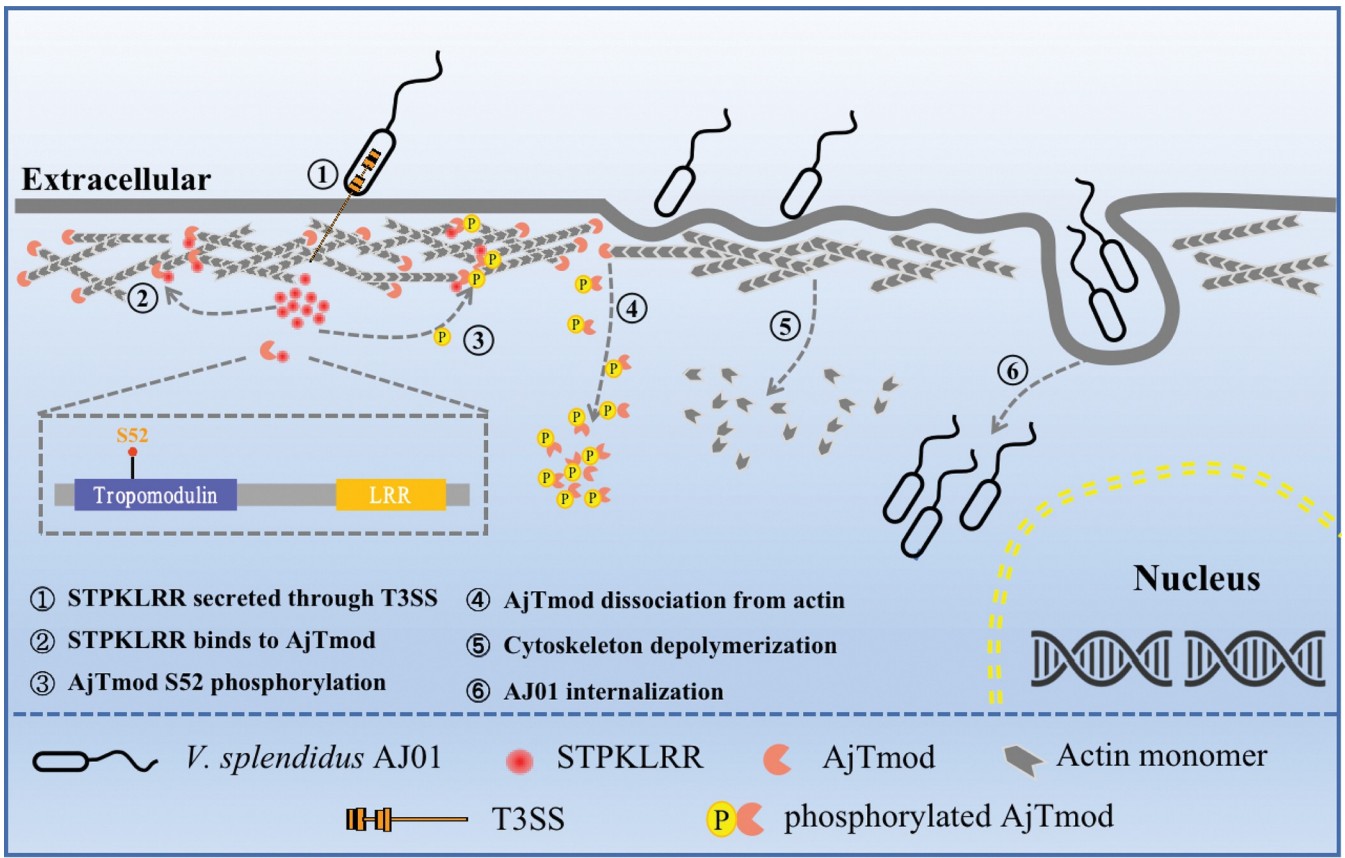

**Fig 8. Schematic of the role of STPKLRR in promoting AJ01 internalization by binding and phosphorylating AjTmod.** As a secreted factor of AJ01 T3SS system (①), STPKLRR was injected into *A. japonicus* coelomocytes and bound to AjTmod (②) to phosphorylate AjTmod at S52 (③). The phosphorylated AjTmod then dissociated from actin and translocated to the cytoplasm (④), which led to actin cytoskeletal rearrangement by changing the F-actin/G-actin ratio (⑤) and ultimately promoting AJ01 internalization (⑥).

D and IPA-3 treatment, respectively, whereas the phagocytosis rate under ΔSTPKLRR infection did not change significantly (Fig 7G).

The change in the F-actin/G-actin ratio is the main mechanism through which actin regulates phagocytosis [35]. Therefore, the F-actin/G-actin ratio was assayed under different conditions. We found that the F-actin/G-actin ratio gradually decreased after AjTmod knockdown by siRNA transfection (Fig 7H, Top strip). In the AJ01 and ΔSTPKLRR::pMP-STPKLRR infection groups, the F-actin/G-actin ratio also gradually decreased with the prolongation of infection time (Fig 7H). However, the F-actin/G-actin ratio in the ΔSTPKLRR and ΔSTPKLRR::pMP2444 groups showed no significant degree of change (Fig 7I).

## Discussion

Host cell invasion is crucial for the pathogenicity and replication of facultative bacterium [36]. In model bacteria, a number of effectors have been confirmed to participate in internalization by targeting the cytoskeleton [37]. In this study, STPKLRR, a novel T3SS effector protein with LRR and STYKc domains, was identified in *V. splendidus* AJ01, a major pathogen of cultured sea cucumbers. This effector was firstly demonstrated to interact with AjTmod, an actin-binding protein in the stabilized cytoskeleton. Mechanistically, STPKLRR acted as a secreted virulence factor to phosphorylate AjTmod at S52 and further promoted the dissociation between

AjTmod and actin. Under this condition, AJ01 was internalized by coelomocytes by effecting actin cytoskeletal rearrangement through changing the F-actin/G-actin ratio. Our results suggested that marine pathogens might employ specific effector molecules to regulate internalization and provided a basis for understanding the diversity of pathogenic mechanisms and candidate targets for developing disease blockades.

Tmod, the only known capping protein for actin, is an important component of the actin cytoskeleton [14, 38]. It has been shown to bind specifically to the pointed ends of actin filaments, thus inhibiting the polymerization and depolymerization of actin monomers [39]. Therefore, targeting Tmod is considered as an effective way to modulate host cell phagocytosis for the internalization of intracellular or facultative bacteria. In this study, STPKLRR, a novel effector targeting AjTmod, which contains a STYKc domain located at 203–403 aa and a LRR domain located at 35–168 aa, was identified in this study. This effector is similar to traditional bacterial serine/threonine protein kinases (STPKs) in other prokaryotic organisms [40], including pathogens, such as *Streptococcus pyogenes* [41], *Mycobacterium tuberculosis* [42], *Yersinia* spp. [43], *L. monocytogenes* [44], and *Pseudomonas aeruginosa* [45]. Some other protein kinases that are homologous to STPKLRR with the conserved motifs and phosphorylation function have also been identified in *A. japonica*, such as protein kinase C Zeta (PKCζ) (GenBank: XP_030828326.1) and protein kinase B (Akt) (GenBank: ATY93171.1). Notably, STPKs share conserved regulatory regions resembling the regions involved in the activation of eukaryotic serine/threonine kinases [46]. Bacterial STPKs and eukaryotic serine/threonine kinases are highly structurally similar mainly because their kinase domains share specific conserved motifs [47]. In STPKLRR, a conserved lysine and the motif "VLGXG" are present in the ATPase active site of the kinase domain, which is used for positioning the phosphate donor ATP molecule and the protein substrate for catalysis [48]. Functionally, in *Streptococcus mutans*, STPKs act as a virulence factor to modulate biofilm formation [49]. Consistent with this previous finding, we confirmed that STPKLRR is a novel virulence factor that destroys host tissue structure via promoting AJ01 internalization (Fig 8).

Protein phosphorylation by STPKs is a key event in signal transduction pathways [48]. Previous study demonstrated that in 3T3-L1 adipocytes, Tmod3 acts as a substrate of Akt2 to regulate actin remodeling under insulin stimulation. S71 of Tmod3 is phosphorylated during insulin-stimulated Akt2 activation, and the phosphorylation of S71 is required for insulin-stimulated glucose transporter (GLUT4) insertion and glucose uptake [50]. The S71 site of Tmod3 is special. It is absent from Tmod1, Tmod2, and Tmod4, as well as from AjTmod. The S52 site of AjTmod and the S71 site of Tmod3 are located between 48 and 92 aa and is the main region of Tmod binding to actin [19]. AjTmod dissociation caused by phosphorylation at S52 is very similar to the actin remodeling effect caused by the phosphorylation of S71 in Tmod3. STPKLRR is a pathogenic protein kinase and shares the same STYKc kinase domain with Akt2. The STYKc domain in bacteria was first discovered in *Myxococcus xanthus* [51], and evidence indicates a close association between its phosphorylation and bacterial pathogenesis [52]. InlA, the internalin of *L. monocytogenes*, promotes adhesion and internalization through an actin-mediated process [53]. InlA, through its leucine repeat regions, binds to the host receptor E-cadherin, promoting its phosphorylation and bacterial uptake [54, 55]. The internalization of AJ01 also requires the phosphorylation of AjTmod at S52, and STPKLRR likely binds to AjTmod through its leucine repeat regions. Moreover, in *M. tuberculosis*, the eukaryotic serine/threonine protein kinase PknG is secreted into host macrophages by blocking the transition of Rab7l1-GDP into Rab7l1-GTP in a kinase activity-dependent process, thus realizing pathogenic potential by facilitating bacterial survival inside human macrophages [56]. The striking similarity of the two-lobed STYKc among PknG, STPKLRR and the eukaryotic Akt2 suggests that bacterial and eukaryotic STYKc share conserved ATP-binding

mechanisms [57]. The N-terminal extremity of the catalytic domain contains a glycine-rich stretch of residues in the vicinity of a lysine residue, which has been shown to be involved in ATP binding [58]. We found that the 209–212 aa sequence "LGQG" in the STYKc domain of STPKLRR is highly conserved with the 96–99 aa sequence "LGKG" of Akt2. This sequence is presumably the ATP-binding site of STPKLRR. The genetic homology of STPKLRR to eukaryotic kinases provides evidence for its function of AjTmod phosphorylation in coelomocytes and suggests that marine pathogens have also evolved means to interfere with host phosphorylation signal transduction.

In summary, we identified a eukaryote-like serine/threonine protein kinase that mediates the internalization of AJ01. These findings provide insight into the mechanisms underlying the internalization of *V. splendidus* and advance our knowledge of the general biology of pathogen-host interactions. Eukaryotic protein kinases are currently the largest group of drug targets, second only to G-protein-coupled receptors, which have drawn attention to eukaryotic-like serine/threonine protein kinases in bacteria [2]. Many STPK inhibitors have been approved by the Food and Drug Administration for use in humans [59], and over 150 kinase inhibitors are also being tested in clinical trials [60]. The STPKLRR from *V. splendidus* AJ01 identified in this study would provide new targets for insights into disease control caused by marine pathogens.

AJ01 internalization is only the first step in bacterial infection, and its subsequent intracellular survival and reproduction are also major challenges. At present, considerable progress has been made in the related research on various other pathogenic bacteria. For example, *M. tuberculosis* has been found to be capable of regulating host intracellular cytosolic nucleic acid-sensing pathways, membrane trafficking and integrity, cell death, and autophagy and other pathways to escape host immunity [61]. *L. monocytogenes* has been demonstrated to modify the phagosome through the effectors listeriolysin O and phospholipase C to escape [62]. STPKLRR or other effectors of intracellular AJ01 involved in this process are largely unknown. How AJ01 escapes phagosomes or disturbs host cell signals needs further investigation.

## Material and methods

### Ethics statement

The sea cucumbers used in this work were commercially bred animals, and all experiments were conducted in accordance with the recommendations of the Guide for the Care and Use of Laboratory Animals of the National Institutes of Health. The study protocol was approved by the Experimental Animal Ethics Committee of Ningbo University, China.

### Animal and coelomocyte culture

Healthy adult sea cucumbers (150 ± 14 g) were collected from the Dalian Pacific Aquaculture Company and acclimatized in seawater (salinity, 28; temperature, 16°C) for 3 days. Sea cucumbers from the experimental and control groups were dissected with sterilized scissors on ice, and their coelomic fluids were filtered through a 300-mesh cell cribble and centrifuged at 800 × *g* for 10 min to harvest coelomocytes for subsequent gene and protein expression analysis.

Primary coelomocytes were prepared according to our previous work [63]. The harvested coelomocytes at a final concentration of $10^6$ cells/mL were cultured at 28°C in L-15 medium with 10 U/mL penicillin, 100 μg/mL streptomycin, and 50 μg/mL gentamicin with osmotic pressure of 0.39 M.

## Bacterial strains, mutagenesis, and complementation

*Escherichia coli* DH5α, S17 λpir, and BL21 (DE3) strains were purchased from TransGen Bio-tech (China) and cultured in Luria-Bertani medium at 37°C. The wild-type *V. splendidus* strain AJ01 was grown in 2216E medium at 28°C. The medium consisted of 5 g/L tryptone, 1 g/L yeast extract (Solarbio, China), and 0.01 g/L FePO$_4$ in filtered seawater.

For the construction of the *STPKLRR* mutant strain, an in-frame deletion mutation of *STPKLRR* was generated via sacB-based allelic exchange as previously described [64]. The upstream and downstream fragments of *STPKLRR* were fused through overlapping PCR with the primer pairs koSTPKLRR-F1/R1 and koSTPKLRR-F2/R2. A 600 bp fragment containing the upstream and downstream regions of *STPKLRR* was cloned into the sacB suicide vector pDM4 and linearized with *Bgl* II, and the correct plasmid was introduced into *E. coli* S17 λpir. Through bacterial conjugation, the resulting plasmid pDM4-ΔSTPKLRR was transformed into wild-type AJ01. The conjugants were placed on 2216E agar with chloramphenicol and ampicil-lin, and the positive conjugants were confirmed through PCR with the primer pairs koSTPKLRR-F1 and koSTPKLRR-R2 and sequenced. The second cross-over recombination process was carried out in 2216E agar with 10% sucrose, and the mutants were verified through PCR with the primer pairs 16s-F and 16s-R and sequenced.

The shuttle plasmid pMP2444 was used to complement ΔSTPKLRR with the *STPKLRR* sequence. The fragment of STPKLRR was inserted into the pMP2444 vector and linearized with *Sal* I and *BamH* I, and the correct plasmid was introduced into *E. coli* S17 λpir. Through bacterial conjugation, the empty plasmid pMP2444 and plasmid with the *STPKLRR* fragment pMP-STPKLRR were transformed into the ΔSTPKLRR strain. The control strain ΔSTPKLRR::pMP2444 and complement strain ΔSTPKLRR::pMP-STPKLRR were verified by using PCR then sequenced.

## Pull-down assay

A pull-down assay was performed to select the AjTmod potential effector from AJ01. Purified recombinant GSTTmod was immobilized on GST-Sefinose Resin (Sangon, China), incubated on ice for 30 min, and washed with wash buffer (pH = 7.3–7.5) for three times. Then, 500 μg of the total resolved protein from AJ01 was added to the above GSTTmod resin and incubated for another 4 h at 4°C. The resin was washed ten times with wash buffer to remove protein impurities and finally eluted with elution buffer (pH = 7.9–8.1). Different truncated AjTmods, including GSTTro and GSTTLRR, were also generated to verify their STPKLRR binding region. The GST tag was used as a control. The eluent was detected by using SDS-PAGE, and differential proteins were characterized by mass spectrometry. The pull-down assay was also performed to screen the AjTmod interactive protein from sea cucumber coelomocytes with 500 μg of total protein as the load with similar procedure.

## Growth curve measurement

The growth curves of four AJ01 strains were assayed in accordance with our previous method [64]. The AJ01, ΔSTPKLRR, ΔSTPKLRR::pMP2444, and ΔSTPKLRR::pMP-STPKLRR strains were plated in 2216E agar at 28°C overnight. A single clone was inoculated into tubes with 10 mL of fresh 2216E medium and cultured at 28°C with shaking at 200 rpm. Overnight cultures were diluted to the same concentration, and 200 μL aliquots of each strain were transferred into flasks with 100 mL of fresh 2216E medium. The flasks were incubated at 28°C with shak-ing at 200 rpm. The optical density at 600 nm (OD$_{600}$) was measured at different time points. Three independent experiments were performed.

## Mortality rate assessment

Sea cucumbers (n = 30) were infected through immersion in seawater containing $10^7$ CFU/mL AJ01, ΔSTPKLRR, ΔSTPKLRR::pMP2444, or ΔSTPKLRR::pMP-STPKLRR in different tanks. The numbers of surviving sea cucumbers in all groups were recorded for 7 days to quantify mortality rates.

## Immunofluorescence

The primary coelomocytes from the experimental and control groups were collected at different time points then fixed with 4% paraformaldehyde for 30 min. Coelomocytes were permeabilized with 0.5% Triton X-100 for 20 min, blocked with 5% BSA in PBST for 30 min, then incubated with polyclonal antibodies diluted to 1:400 in 5% PBST at 37°C for 1 h. After washing the slide three times with PBST for 10 min, the membrane was subsequently incubated with the secondary antibody diluted to 1:800 at 37°C for 1 h. The slide was washed and stained with DAPI (Beyotime, China) and sealed with antifade mounting medium (Beyotime, China). The slide was visualized by using a laser scanning spectral confocal microscope (TCS SP2; Leica, Solms, Germany). For confirming the colocalization of two proteins in accordance with the method reported by Chai et al. [65], a line was first drawn along the boundary of the cell to fully surround the cell. Then, the integrated intensity of each channel of interest was determined by using ImageJ. The fluorescence intensity of each fluorescent protein was determined with the Plot Profile tool.

## STPKLRR translocation by β-lactamase system assay

The translocation of STPKLRR was evaluated in infected coelomocytes via β-lactamase system as previously described by Chen et al. [66]. Briefly, the pCX340 control vector and STPKLRR fused vector pCX340-STPKLRR were introduced into AJ01 through bacterial conjugation, respectively. Coelomocytes were infected with control and experimental AJ01 at a MOI of 100. After 1, 3, 6, and 12 h infection, the harvested coelomocytes (approximately $10^6$ cells/mL) were centrifuged at $400 \times g$ for 10 min and washed three times with fresh 2216E medium. Then, the coelomocytes were resuspended in 250 μL β-lactamase loading solution containing 0.125 μg the fluorescent substrate CCF4-AM (LiveBLAzer-FRET B/G loading kit; Invitrogen) and 15 mM Probenecid (Invitrogen). The coelomocytes were subsequently incubated in the dark for 120 min at room temperature before confocal microscope observation.

## AjTmod mutagenesis

Five AjTmod serine mutants were constructed in accordance with a previous method [67]. By using the *Fast* Mutagenesis System (TransGen Biotech, China), five serine sites in the tropomodulin domain of AjTmod were mutated into alanine, respectively. In brief, PCR amplification was performed with pGEX-4T-2-Tmod as the template and the mutation primers listed in S3 Data. Subsequently, the PCR product was added with 1 μL of DMT enzyme and incubated at 37°C for 1 h before transformation into DMT competent cells. The positive mutants were validated by sequencing, and protein expression was induced by IPTG. The mutants were named as S12A, S42A, S49A, S52A, and S144A, respectively.

## In vitro protein kinase activities assay

*In vitro* kinase activities assays were performed on purified GSTTmod and five mutant proteins. In brief, 1 μg of GST-Tmod and 50 ng of His-tagged STPKLRR were incubated in 50 μL kinase buffer (25 mM Tris-Cl, pH 7.5, 5 mM β-glycerophosphate, 2 mM dithiothreitol [DTT],

0.1 mM $Na_3VO_4$, and 10 mM $MgCl_2$) containing 0.2 mM ATP. The kinase reaction system was incubated at 30°C for 30 min and separated via SDS-PAGE. AjTmod phosphorylation was detected by using Phos-tag Acrylamide AAL-107 (Wako, Japan). Samples were analyzed through SDS-PAGE and Western blot.

## Western blot analysis

A total of 50 μg of protein from each sample was separated through SDS-PAGE then transferred to a 0.45 mm pore nitrocellulose membrane with an Enhanced ChemiLuminescence (ECL) Semidry Blotter (Amersham Biosciences, USA). The membrane was blocked with 5% skim milk in TBST (20 mM Tris-HCl, 150 mM NaCl, and 0.05% Tween-20) at 37°C for 1 h. The membranes were then incubated with polyclonal antibodies diluted to 1:400 in 5% skimmed milk at 4°C for 12 h. After washing the membrane for three times with TBST for 10 min, the membrane was subsequently incubated with diluted goat-anti mouse or goat-anti rabbit IgG (Sangon, China) diluted to 1:3000 at room temperature for 1 h. The membrane was further washed with TBST for three times and incubated in Western Lightning-ECL substrate (Perkin Elmer, USA) prior to exposure to a X-OMAT AR X-ray film (Eastman Kodak, USA). Band density was analyzed with ImageJ, and protein expression levels were normalized to that of β-Tubulin control.

All antibodies used in this study could be found in the key resources table, Table 1. STPKLRR, Hop, FliC, and Tmod antibodies were generated in accordance with a previously described method [68, 69]. The Hop, FliC, and Tmod antibodies had been successfully used in our previous works [18, 70, 71]. The specificity of the STPKLRR antibody was shown in Fig 3A.

## Coimmunoprecipitation assay

For coimmunoprecipitation, 1 mL coelomocytes ($10^6$ cells/mL) treated under different conditions were lysed on ice by using 100 μL RIPA lysis buffer for 30 min (50 mM Tris [pH 7.4], 150 mM NaCl, 1% NP-40, 0.5% sodium deoxycholate, and 0.1% SDS). The supernatant was collected and immunoblotted with 1 μg either AjTmod or actin antibodies for 4 h. Then, protein A+G agarose (Beyotime, China) washed three times with 30 μL RIPA lysis buffer was added and incubated for another 4 h. After that, the beads were collected and boiled with 5 × loading buffer for 10 min. The proteins were detected through Western blot analysis.

## Microscale thermophoresis (MST) assay

MST assay was performed on the basis of our previous work with some modifications [18]. The affinity of the purified GSTTmod to His-fused actin-binding protein was quantified by using Monolith NT.115 (Nanotemper Technologies). GSTTmod and His-fused actin were fluorescently labeled in accordance with the manufacturer's procedure. A total of 100 μL of 10 μM protein solution of GSTTmod and His-fused actin was exchanged with labeling buffer. Then, the sample was mixed with 300 μL of the fluorescent dye NT-647-NHS and incubated for 30 min at 25 °C in the dark. Finally, the labeled proteins were dialyzed with column B (Nanotemper L001) and eluted with 50 mM Tris-HCl (pH 8.0) supplemented with 0.02% Tween 20. For the MST assay, the labeled protein (approximately 5 μM) was incubated with the same volume of unlabeled His-fused actin/His tag/GST tag of 16 different serial concentrations in 50 mM Tris-HCl (pH 8.0) supplemented with 0.02% Tween 20 at room temperature for 10 min. Then, two tubes of GSTTmod/His-fused actin were added to His-fused STPKLRR and His tag (5 μM) and incubated for 10 min. The samples were then loaded into silica

**Table 1. Antibodies, bacterial strains, chemicals, plasmids used in this study.**

| REAGENT or RESOURCE | SOURCE | IDENTIFIER |
|---|---|---|
| **Antibodies** | | |
| GST-Tag(12G8) mouse antibody | Abmart | Cat# M20007; RRID:AB_2864360 |
| His-Tag mouse antibody | Proteintech | Cat# CL647-66005; RRID:AB_2920271 |
| ATP1A1 polyclonal antibody | Proteintech | Cat# 14418-1-AP; RRID:AB_2227873 |
| DnaK antibody | Abmart | Cat# PH3459 |
| OmpK antibody | Abmart | Cat# PH9512 |
| Beta Tubulin antibody | Proteintech | Cat# 10068-1-AP; RRID:AB_2303998 |
| Beta Actin antibody | Proteintech | Cat# 66009-1-Ig; RRID:AB_2687938 |
| Goat Anti-Rabbit IgG (H&L)-Alexa Fluor 647 | Abmart | Cat# M213811 |
| CoraLite488-conjugated Goat Anti-Mouse IgG | Proteintech | Cat# SA00013-1 |
| Cy3–conjugated Affinipure Goat Anti-Rabbit IgG | Proteintech | Cat# SA00009-2 |
| Anti-E. coli RNA Polymerase β Antibody | Biolegend | Cat# 663903 |
| STPKLRR | This paper | N/A |
| Hop | This paper | N/A |
| Tmod | This paper | N/A |
| FliC | This paper | N/A |
| **Bacterial strains** | | |
| *Vibrio splendidus* AJ01 | This paper | N/A |
| *Vibrio splendidus* ΔSTPKLRR | This paper | N/A |
| *Vibrio splendidus* ΔSTPKLRR::pMP2444 | This paper | N/A |
| *Vibrio splendidus* ΔSTPKLRR::pMPSTPKLRR | This paper | N/A |
| **Chemicals, and recombinant proteins** | | |
| CCF4-AM | Thermofisher | Cat# K1028 |
| Cinnamaldehyde | Merck | Cat# W228613; CAS: 104-55-2 |
| Licoflavonol | MedChemExpress | Cat# HY-N6583; CAS: 60197-60-6 |
| Actin-Tracker Red-555 | Beyotime | Cat# C2203S |
| Dil | Beyotime | Cat# C1036 |
| DAPI | Beyotime | Cat# C1002 |
| Ponceau S | Abmart | Cat# A10010 |
| DTAF | Merck | Cat# D0351; CAS: 21811-74-5 |
| Chlorpromazine | Sangon Biotech | Cat# A506232; CAS: 69-09-0 |
| Nystatin | Sangon Biotech | Cat# A600390; CAS: 1400-61-9 |
| IPA-3 | MedChemExpress | Cat# HY-15663; CAS: 42521-82-4 |
| Cytochalasin D | MedChemExpress | Cat# HY-N6682; CAS: 22144-77-0 |
| Mitmab | Selleck | Cat# E1081; CAS: 2253617-91-1 |
| rAjTmod and the mutant | This paper | N/A |
| rSTPKLRR | This paper | N/A |
| **Critical commercial assays** | | |
| LC-MS/MS | Sangon Biotech | N/A |
| Phos-tag™ Acrylamide AAL-107 | Wako | Cat# 300–93523 |
| HOOK Cell Surface Protein Isolation Kit | Sangon Biotech | Cat# C006316 |
| Pierce BCA Protein Assay Kit | Life Technologies | Cat# 23225 |
| Human Actin Elisa Kit | Sinovac Biotech | Cat# F7230-A |
| **Deposited data** | | |
| LC-MS/MS results are listed in S1 and S2 Data files | This paper | N/A |
| **Experimental models: Organisms/strains** | | |
| Apostichopus japonicus | This paper | N/A |

(*Continued*)

**Table 1.** (Continued)

| REAGENT or RESOURCE | SOURCE | IDENTIFIER |
|---|---|---|
| Escherichia coli BL21 | TransGen Biotech | Cat# CD901 |
| Escherichia coli Transetta (DE3) | TransGen Biotech | Cat# CD801 |
| **Oligonucleotides** | | |
| Primers are listed in S3 Data | This paper | N/A |
| **Recombinant DNA** | | |
| pGEX-4T-2-AjTmod (or the mutants) | This paper | N/A |
| pGEX-4T-2-AjTTro | This paper | N/A |
| pGEX-4T-2-AjTLRR | This paper | N/A |
| pET28a-STPKLRR | This paper | N/A |
| pDM4-ΔSTPKLRR | This paper | N/A |
| pMP2444-STPKLRR | This paper | N/A |
| pCX340-STPKLRR | This paper | N/A |
| **Software and algorithms** | | |
| GraphPad Prism version 6 | GraphPad Software | RRID:SCR_002798, URL: https://www.graphpad.com/ |
| ZEN | Zeiss | https://www.zeiss.com/microscopy/us/products/microscope-software.html |
| ImageJ | NIH, USA | https://imagej.nih.gov/ij |

capillaries (Polymicro Technologies) and measured at 25 ˚C at 20%–40% LED power and 20% MST power. Data analyses were performed by using the Nanotemper analysis software.

## ELISA analysis

The 96-well plates used for ELISA were blocked with 5% BSA in PBS at 37˚C for 3 h. After being washed three times with PBS containing 0.05% Tween 20 (PBST), 0.1, 0.2, 0.4, 0.8, and 1.6 μg of His-fused actin protein were added to 100 μL of ELISA coating buffer (Solarbio, China) and incubated overnight. After being washed three times with PBST, 1 μg of recombinant GSTTmod and the mutants S12A, S42A, S49A, S52A, and S144A were added to each well and maintained for 3 h. The plate was washed three times with PBST. Then, 100 μL of the diluted AjTmod antibody (1:1000 in PBS) was added to each well and incubated at 37˚C for 1 h. After washing, the wells were treated with 1:3000 diluted goat-anti mouse IgG (Sangon, China) at 37˚C for 1 h. After the last three washes, a TMB Kit (Solarbio, China) was used for color development, and 50 μL of hydrochloric acid (1 M) was added to terminate the reaction. The absorbance of the developed color was read at 450 nm with a UV-Vis spectrophotometer (Beckman, USA).

## RNA silencing

The specific siRNAs for the target and control genes were synthesized by GenePharma (China). The detailed sequence information of siAjTmod and siNC is shown in S3 Data. The experimental and control siRNAs were dissolved in RNase-free water to obtain 20 μM working solutions. A total of 10 μL of siRNA, 10 μL of Lipo6000 transfection reagent (Beyotime, China), and 80 μL of PBS were mixed and used as the transfection solution. Sea cucumbers were injected with 100 μL of the transfection solution of siNC or siAjTmod groups. After transfection for 3, 6, 12, and 24 h, coelomocyte samples were collected and used to detect interference efficiency.

## AJ01 internalization assay in vitro

The bacterial internalization assay was performed in accordance with a previously reported method [72]. The log-phase grown AJ01 were then inoculated to $10^6$ coelomocytes in six-well plates at MOI = 10. At 1, 2, and 3 h, infected coelomocytes were treated with 100 μg/ml gentamycin for 2 h to kill extracellular bacteria. The collected coelomocytes were then washed thoroughly with PBS to remove the bacteria remaining in supernatant, then further incubated for 30 min. Next, coelomocytes containing intracellular bacteria were extracted by using 0.2% TritonX-100 used for intracellular bacterial counting and Western blot analysis. The collected intracellular bacteria were diluted in a gradient, plated on 2216E solid medium, and counted. The colonies were subjected to 16S rDNA sequencing.

## Determination of phagocytosis rate via flow cytometry

*V. splendidus* AJ01 was stained with 5-DTAF (Sigma, USA) as previously described [73]. The working concentration of 5-DTAF was 0.005 mg/mL (dissolved in PBS). A total of 5 mL of mid-logarithmic phase AJ01 ($OD_{600}$ = 0.5) and 2 mL of the 5-DTAF solution were mixed in a dark environment at 28°C and incubated in an oscillating incubator for 1 h. Then, DTAF-labeled AJ01 (DTAF-AJ01) cells were collected through centrifugation at $8000 \times g$ for 6 min and resuspended in PBS. The wells were repeatedly washed until the supernatant became colorless. Subsequently, DTAF-labeled AJ01 was used for the sea cucumber injection experiment. Coelomocytes were harvested at 3 h after AjTmod knockdown or treatment with different inhibitors, and phagocytic activity was detected by using flow cytometry (ImageStreamX MarkII, USA). A total of 10000 cells were acquired to quantify the percentage of phagocytic activity. The experiments were performed with three independent replicates.

## Assay of the AjTmod-mediated endocytic pathway under inhibitor treatment

Five specific phagocytosis inhibitors, namely, CPZ (Sangon Biotech), nystatin (Sangon Biotech), IPA-3 (Aladdin Shanghai), cytochalasin D (Aladdin Shanghai), and mitmab (Aladdin Shanghai), were used for the endocytic inhibition assay in accordance with our previous work [34]. AjTmod was knocked down via specific siRNA transfection *in vivo* to explore the effect of AjTmod on phagocytosis. Subsequently, DTAF-labeled AJ01 ($10^6$ CFU/mL) was injected into sea cucumbers for another 3 h, and phagocytic activity was determined through flow cytometry. Different inhibitors were added to AjTmod-silenced cultured coelomocytes to investigate the AjTmod-mediated endocytosis pathway. After DTAF-AJ01 injection for another 3 h, the phagocytic activity was determined via flow cytometry.

## Determination of the F-actin/G-actin ratio

The F-actin/G-actin ratio was determined in accordance with a previously reported method [74]. Coelomocytes were collected at 3, 6, 12, and 24 h after AJ01 infection. Equal numbers of coelomocytes from each time point were lysed in cold lysis buffer for 30 min (10 mM $K_2HPO_4$, 100 mM NaF, 50 mM KCl, 2 mM $MgCl_2$, 1 mM EGTA, 0.2 mM DTT, 0.5% Triton X-100, 1 mM sucrose, pH 7.0) and centrifuged at $15\,000 \times g$. The amount of soluble actin (G-actin) in the supernatant was measured. The insoluble F-actin in the pellet was resuspended in lysis buffer plus an equal volume of buffer 2 (1.5 mM guanidine hydrochloride, 1 mM sodium acetate, 1 mM $CaCl_2$, 1 mM ATP, 20 mM TrisHCl, pH 7.5) and incubated on ice for 1 h with gentle mixing every 15 min to convert insoluble F-actin into soluble G-actin. The samples were centrifuged at $15 \times 000\,g$ for 30 min, and the amount of F-actin in the supernatant was

measured. Then, the F-actin and G-actin fractions were analyzed through Western blot analysis.

## Statistical analysis

Statistical analyses were performed by using GraphPad Prism (GraphPad Software). Standard deviation was used to reflect the dispersion of multiple independent variables relative to the mean. Unpaired two-tailed Student's t-tests were conducted for the single comparison of two groups. Two-way analysis of variance (ANOVA) was used for analysis of experiments with multiple groups and multiple independent variables, and one-way ANOVA was used for analysis of multiple groups with a single independent variable. All data are representative of at least three independent experiments and presented as the mean ± SD. Statistical significance was defined as $p > 0.05$, not significant (n.s.); *$p < 0.05$; **$p < 0.01$; and ***$p < 0.001$. The numerical data used in all figures are included in S4 Data.

## Supporting information

**S1 Fig. For the identification of effectors targeting AjTmod from AJ01, a pull-down assay was performed with recombinant GSTTmod (GST-fused complete AjTmod) as the bait and AJ01 total protein as the load. (A)** Differential bands were detected by using SDS-PAGE. **(B)** The differential bands were further characterized through mass spectrometry, and the top 20 AjTmod-interacting proteins are listed.
(TIF)

**S2 Fig. Knockdown and complement of STPKLRR. (A)** Construction of the STPKLRR mutant strain ΔSTPKLRR. Left panel: The STPKLRR mutant was successfully constructed by two homologous recombination through bacterial conjugation. Right panel: Validation of STPKLRR knockdown by western blotting. **(B)** Construction of the complemented strain ΔSTPKLRR::pMP-STPKLRR, and the control complemented strain ΔSTPKLRR::pMP2444. Left panel: The plasmid pMP/STPKLRR and the empty plasmid pMP2444 were successfully transferred into the STPKLRR mutant strain ΔSTPKLRR through bacterial conjugation. Right panel: Validation of STPKLRR complement by western blotting.
(TIF)

**S3 Fig. STPKLRR is an essential virulence factor for AJ01 infection. (A)** No significant growth difference was detected among AJ01, ΔSTPKLRR, ΔSTPKLRR::pMP2444, and ΔSTPKLRR::pMP/STPKLRR. **(B)** Muscles, intestines were subjected to histological observation at 72 h after ΔSTPKLRR::pMP2444 and ΔSTPKLRR::pMPSTPKLRR infection to further confirm the pathogenic effect. Black arrows represent areas of tissue damage. Scale bar, 100 μm. **(C)** Intracellular ΔSTPKLRR::pMP2444 and ΔSTPKLRR::pMPSTPKLRR (red arrows) was detected by transmission electron microscopy.
(TIF)

**S4 Fig.** qRT-PCR **(A)** and western blotting analysis **(B)** of STPKLRR expression in sea cucumber coelom fluid treatment. RNAP, RNA Polymerase, was used as a bacterial cytosolic marker. The data, which are presented as the means ± SDs (n = 3) relative to the negative group, are shown in bar graphs, respectively. Asterisks indicate significant differences compared with the control group: ***$p < 0.001$ and **$p < 0.01$ (t-test). **(C)** Intracellular AJ01 burden was collected and plated on 2216E solid medium with gradient dilution (left panel). The single colonies on the 2216E solid medium were counted (right panel).
(TIF)

**S5 Fig. Screening of AJ01 T3SS Inhibitors.** **(A)** Western blot analysis of the specificity of Hop antibody. The optimal concentration of Salicylidene acylhydrazide **(B)**, Phenoxyacetamide **(C)**, Piericidin A **(D)**, Quercetin **(E)** and Quinine **(F)**. The absorbance value of AJ01 with different concentrations of inhibitors was measured at 600 nm at each hour. The optimal inhibitor concentration that does not affect the growth of AJ01, was Salicylidene acylhydrazide 50 μM, Phenoxyacetamide 100 μM, Piericidin A 50 μM, Quercetin 200 μM, Quinine 50 μM, respectively. **(G-K)** After 16 h treatment of each inhibitor, the secreted proteins of AJ01 were extracted for western blotting. DnaK, a marker of bacterial cytosolic marker; Hop, a marker of vibrio T3SS secreted proteins. Band density was quantified using ImageJ, and protein levels of Hop and STPKLRR treated inhibitors were quantified and normalized to the NC group. Data (means ± SD) are representative of at least 3 experiments. Asterisks indicate significant differences (*$p < 0.05$; ***$p < 0.001$). (TIF)

**S6 Fig. Effect of STPKLRR on AjTmod phosphorylation and coelomocytes phagocytosis.** **(A)** Coelomocytes treated with 100 μg STPKLRR or His tag for 12 h were collected and used for the detection of AjTmod phosphorylation. **(B)** After treated with 100 μg STPKLRR or His tag for 12 h, 20 μL FITC-labeled fluorescent microspheres were added and incubated for another 3 h. **(C)** The coelomocytes phagocytosis rate of fluorescent microspheres was detected by flow cytometry. Data (means ± SD) are representative of at least 3 experiments. Asterisks indicate significant differences (***$p < 0.001$). (TIF)

**S7 Fig. Functional assay of AjTmod-actin affinity to actin cytoskeleton.** Coelomocytes were infected with AJ01 or ΔSTPKLRR (MOI = 100) for 24 h and collected for the immunofluorescence (**A**, Scale bar, 5 μm) and transmission electron microscope analysis (**B**, Scale bar, 2 μm, and 1 μm). (TIF)

**S8 Fig. AjTmod modulates coelomocyte phagocytosis.** AjTmod-interacting proteins were identified by using a pull-down assay and further characterized by mass spectrometry to determine AjTmod-mediated processes in coelomocytes. The identified proteins were enriched in GO **(A)**, KEGG **(B)**, and PPI **(C)** analyses. (TIF)

**S9 Fig. The lysosome activity of coelomocytes infected with AJ01 and ΔSTPKLRR.** After AJ01 and ΔSTPKLRR 0 h, 12 h and 24 h infection, coelomocytes were collected, incubated with Lyso-Tracker Red to label the lysosomes, stained with DAPI and then observed under a laser-scanning confocal microscope. Scale bar, 5 μm. (TIF)

**S1 Data. LC-MS_MS results of AjTmod-interacting Vibrio splendidus proteins.** (XLSX)

**S2 Data. LC-MS_MS results of AjTmod-interacting coelomocyte proteins.** (XLSX)

**S3 Data. Primers.** (XLSX)

**S4 Data. The Numerical data used in all figures.** (XLSX)

## Acknowledgments

We would like to thank Qiyao Wang from East China University of Science and Technology for providing the pCX340 plasmid.

## Author Contributions

**Conceptualization:** Fa Dai, Chenghua Li.

**Formal analysis:** Fa Dai.

**Funding acquisition:** Chenghua Li.

**Investigation:** Fa Dai.

**Methodology:** Fa Dai.

**Project administration:** Chenghua Li.

**Resources:** Chenghua Li.

**Supervision:** Ming Guo, Chenghua Li.

**Writing – original draft:** Fa Dai.

**Writing – review & editing:** Ming Guo, Yina Shao, Chenghua Li.

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
