## [Decision Letter · Decision Letter 0]

3 Apr 2023

Dear Dr. Li,

Thank you very much for submitting your manuscript "Novel secreted STPKLRR from Vibrio splendidus AJ01 promotes pathogen internalization via mediating tropomodulin phosphorylation dependent cytoskeleton rearrangement" for consideration at PLOS Pathogens. As with all papers reviewed by the journal, your manuscript was reviewed by members of the editorial board and by several independent reviewers. In light of the reviews (below this email), we would like to invite the resubmission of a significantly-revised version that takes into account the reviewers' comments.

Reviewer 1 and 4 complain that the multiple figure panels are too small. The Figures all need attention so that it is possible to properly read the text on the Y and X-axis and all photos (Figures) must be enlarged so they can be read and studied without any magnification. There are some experiments proposed especially by Reviewer 1 but also one experiment on lysosomal activity suggested by Reviewer 3 that need to be done unless the authors can convincingly argue that these experiments are not necessary. It is important that all comments made by each reviewer are properly dealt with and if the authors disagree with any recommendation or criticism they have to deliver good arguments for doing this.

We cannot make any decision about publication until we have seen the revised manuscript and your response to the reviewers' comments. Your revised manuscript is also likely to be sent to reviewers for further evaluation.

Sincerely,

Kenneth Söderhäll

Guest Editor

PLOS Pathogens

Karla Satchell

Section Editor

PLOS Pathogens

Kasturi Haldar

Editor-in-Chief

PLOS Pathogens

orcid.org/0000-0001-5065-158X

Michael Malim

Editor-in-Chief

PLOS Pathogens

orcid.org/0000-0002-7699-2064

Reviewer 1 and 4 complain that the multiple figure panels are too small. The Figures all need attention so that it is possible to properly read the text on the Y and X-axis and all photos (Figures) must be enlarged so they can be read and studied without any magnification. There are some experiments proposed especially by Reviewer 1 but also one experiment on lysosomal activity suggested by Reviewer 3 that need to be done unless the authors can convincingly argue that these experiments are not necessary. It is important that all comments made by each reviewer are properly dealt with and if the authors disagree with any recommendation or criticism they have to deliver good arguments for doing this.

Reviewer's Responses to Questions

**Part I - Summary**

Reviewer #1: In this paper the authors identify a leucine-rich repeat-containing serine/threonine protein kinase from Vibrio splendidus AJ01. The authors use several methods to demonstrate that this protein is used by the pathogen to phosphorylate Tropomodulin (Tmod) in Apostichopus japonicus, leading to that Tmod dissociate from actin which result in rearrangement of the cytoskeleton enabling internalization of the pathogenic AJ01 bacterium. This is an important finding in order to understand the infection biology of V. splendidus.

The findings are new and very interesting and is a continuation of previous studies by this group, and the authors have done a proper and thorough coverage of previous relevant studies in the literature.

Reviewer #2: This report is an important contribution to our understanding about the detailed mechanisms used by Vibrio splendidus to enter Apostichopus japonicus coelomocytes. The authors provide clear evidence for the role of the serine/threonine kinase activity of STPKLRR, to phosphorylate tropomodulin in the coelomocyte cytoplasm, in order to favor V. splendidus internalization by phagocytosis most likely through macropinocytosis and actin-dependent endocytic pathways.

Tropomodulin in Apostichopus japonicus was first characterized in the group’s previous work in JBC, which plays an important role in coelomocytes apoptosis. Based on the first publication, the authors identify a Tmod-targeted mechanism of cell entry in Vibrio splendidus with a considerable number of assays. This makes sense from a pathogen-host interaction perspective.

The role of Tmod in stabilizing the cytoskeleton seems to be its own job. However, Tmod plays a role in apoptosis, which is not discussed in the manuscript. The multifunctionality of a single gene is not surprising, but the involvement of pathogen-host interaction at the same time is worthy of attention in the follow-up research of the authors.

However, there are still some details in the manuscript needed to be addressed.

Reviewer #3: This manuscript mainly revealed the mechanisms that a novel Vibrio splendidus AJ01 Type III secretion system effector of STPKLRR could specifically interact with AjTmod, causing the phosphorylation of AjTmod at serine 52, thus reducing the binding stability between AjTmod and actin. After AjTmod dissociated from actin, the F-actin/G-actin ratio decreased to induce cytoskeletal rearrangement, which in turn promoted the internalization of AJ01. Supported by a large number of experimental data, this manuscript answers how AJ01 breaks the AjTmod-stabilized cytoskeleton for internalization from fine details.

Reviewer #4: This is an interesting study which adds to advancing our understanding of TSS3 mechanisms of Vibrio splendidus in the host Apostichopus. The study is comprehensive, with a full range of different advanced methods being employed to provide insights of mechanisms. The general execution and scholarship appear sound. However, the resolution and size (and complexity) of the multi panel figures completely defeats review. Image resolution is so poor - it is impossible to judge the detail of what is being shown, or follow the interpretation or verify claims.

I would advise images are submitted at higher resolution, and would actually encourage the submission of simpler figures with fewer panels per composite figure.

**Part II – Major Issues: Key Experiments Required for Acceptance**

Reviewer #1: The data partly support the conclusions, but some more controls and/or explanations are needed, since some results are confusing. The authors have to explain their results in a clearer way and not use so many panels in each figure. Most of the panels in the figures are far too small and the text is nearly impossible to read, and therefore very difficult to evaluate. This is also true for the some of the supplementary figures. For example, figure S6A-C are not readable so either remove or magnify.

For figure 1D an explanation is needed in the legend about the details of the upper and lower panels. The figure should indicate His-STPKLRR and not only SPKLRR. Moreover, the pattern of the lanes in the blot for His-staining differ a lot from the blots for GST staining. Is the same amount of protein loaded on these gels? Please indicate the loading amount. The panels in figure 1 is also very small making them difficult to read and evaluate. I suggest that the authors place figure 1A-C as a supplement, and enlarge figure 1D.

Figure 2D is to low resolution to be understandable. It is not possible to understand what the arrow points at. There are too many panels in this figure as well. In figure 2G the number of intracellular AJ01 is shown, but to what are these figures related? Is it per gram tissue or something else?

Another question is to the following experiments where coelomocytes are used. Are these cells the main target of this bacterium? In figure 2E second row the bacterium is in a phagosome, but is it killed there or does it multiply in coelomocytes?

Figure 3 has too many panels, which are too small to be readable. Magnification does make the panels blurry. The text in figure 3E-H is impossible to read. Figure 3A-B shows secretion of STPKLRR and uptake (?) into coelomoctes. The legend is a bit unclear. In 3A marker lanes for the western blot is needed, and for 3B the legend should indicate that the pictures show coelomocytes and internalization of STPKLRR (if this is correct).

Figure 3D shows coelomocyte extract from cell infected with the different mutants and “wild type” AJ01 strains used in figure 2, and STPKLRR is only found in the wild type infected cells or the “restored” DeltaSTPKLRR::pMPSTPKLRR infected cells. However, why is DnaK detected in the cytoplasm of all coelomocytes if was necessary for bacterial internalization as shown in figure 2?

Figure 3E-G: Explain that these figures show the effect of the T3SS inhibitors on bacterial growth (if this is correct). These panels could be moved to supplemental material.

The DnaK panels in figure H-J looks completely empty (non loaded), and moreover the non-existent variability of the band intensity shown in the diagrams 3H-J looks very strange. Is it really possible to get nearly identical blots from three different biological replications? What does Gary value mean?

I am a bit concerned over figure 4A (and 4B), since the phosphorylated bands look pasted, and have moved in a straight manner compared to the non-phosphorylated proteins. The other panels in figure 4, except for E, F and L are too small to evaluate. The concentrations used in figure 4K are not visible, and are important to judge the results.

In figure 5H the total amount of actin varies between time points. Is this correct?

Reviewer #2: The morphology of intracellular bacteria in the second and fifth panels of Figure 2E is obviously different. Why is the case? Is this due to the introduction of the vector or other bacteria? If not, some plausible explanation is needed to demonstrate the possible cause of the intracellular morphological changes in Vibrio splendidus.

Reviewer #3: 1. Is STPKLRR located on the surface of AJ01? How does AJ01 expose STPKLRR to bind Tmod during internalization?

2. Does STPKLRR-mediated AJ01 internalization promotes the phagocytosis of AJ01? Phagocytosis is a strategy for eliminating pathogens, but STPKLRR-mediated AJ01 internalization promotes AJ01 infection, which seems to contradict each other in this manuscript. It is recommended to test lysosomal activity.

Reviewer #4: I would not recommend that further experiments are necessary.

**Part III – Minor Issues: Editorial and Data Presentation Modifications**

Reviewer #1: The authors need to improve the quality of all the small panels and arrange the paper in a readable way. The English also needs professional editing.

Reviewer #2: - Line 61: The provided references of MARTX in Vibrio cholerae are wrong. Please provide the correct references.

- Line 77: "Tmod, the only known capping protein of F-actin" is inaccurate. It is the only pointed (minus) end-capping protein.

- Check spellings: β-Tubulin or β-tubulin? It should be unified. In Fig. 5, coleomocytes should be corrected to coelomocytes.

- In Fig. 2 and Fig. 3, there are some panels with small text. They are difficult for a reader to judge.

- Figure S6 can better connect the logic of the manuscript. It is suggested to move it to the main text of the manuscript.

Reviewer #3: 1. Fig 3C, the luminance of DAPI is inconsistent. Is the setting of fluorescence parameters inconsistent? In addition, STPKLRR seems to affect the expression of Tmod, but this experiment is not mentioned in this paper.

2. Fig 4A, lacking of internal reference, so there is no comparability between different experimental groups.

3. Fig4F, it is recommended to add STPKLRR signal.

4. Fig5, although F-actin/G-actin ratio indicates cytoskeleton rearrangement, it is suggested to add immunofluorescence assay to observe cytoskeleton changes.

Reviewer #4: Resolution of figures must be dramatically improved. For example, the histopathology shown in figure 2 is impossible to judge. E.g. figure 4 is too small, and too low resolution to follow. Etc.

PLOS authors have the option to publish the peer review history of their article (what does this mean?). If published, this will include your full peer review and any attached files.

Reviewer #1: No

Reviewer #2: No

Reviewer #3: No

Reviewer #4: No
---

## [Decision Letter · Decision Letter 1]

11 May 2023

Dear Dr. Li,

We are pleased to inform you that your manuscript 'Novel secreted STPKLRR from Vibrio splendidus AJ01 promotes pathogen internalization via mediating tropomodulin phosphorylation dependent cytoskeleton rearrangement' has been provisionally accepted for publication in PLOS Pathogens.

Best regards,

Kenneth Söderhäll

Guest Editor

PLOS Pathogens

Karla Satchell

Section Editor

PLOS Pathogens

Kasturi Haldar

Editor-in-Chief

PLOS Pathogens

orcid.org/0000-0001-5065-158X

Michael Malim

Editor-in-Chief

PLOS Pathogens

orcid.org/0000-0002-7699-2064

Reviewer Comments (if any, and for reference):

Reviewer's Responses to Questions

**Part I - Summary**

Reviewer #1: This is a revised manuscript and the authors have responded to all questions in an accurate way. The paper is very interesting and the authors present new valuable information about Vibrio infections.

Reviewer #2: The manuscript has been revised according to the reviewer's comments.

Reviewer #3: The findings in this study are interesting for the host-pathogens interaction in invertebrates.

Reviewer #4: This revised manuscript has addressed my main (and only) criticism of poor image quality which made initial assessment impossible.

**Part II – Major Issues: Key Experiments Required for Acceptance**

Reviewer #1: None.

Reviewer #2: No

Reviewer #3: The key experiments have been added.

Reviewer #4: None

**Part III – Minor Issues: Editorial and Data Presentation Modifications**

Reviewer #1: The figures have been changed and some panels have been moved to supplement. However, I think still that the figures contain to many small panels, and are very difficult to read without very magnification. Moreover they do not fulfil the requirement from PLoS Pathogens instructions, where the size limit is 2625 pixel width at 300 dpi. The authors have made more than 2,5 as large figures in order to get all small panels into the same figure. Please read instructions and make more clear and readable figures.

For figure 2: Add the explanation for figure 2F in the figure legend: The number of intracellular bacteria represented the number of intracellular bacteria in each well of a 6-well plate, and the number of cells per well was about106.

Reviewer #2: No

Reviewer #3: Ok for this reviewer.

Reviewer #4: None

PLOS authors have the option to publish the peer review history of their article (what does this mean?). If published, this will include your full peer review and any attached files.

Reviewer #1: No

Reviewer #2: No

Reviewer #3: **Yes: **Hai-peng Liu

Reviewer #4: No

---

## [Editor Report · Acceptance letter]

18 May 2023

Dear Dr. Li,

We are delighted to inform you that your manuscript, "Novel secreted STPKLRR from Vibrio splendidus AJ01 promotes pathogen internalization via mediating tropomodulin phosphorylation dependent cytoskeleton rearrangement," has been formally accepted for publication in PLOS Pathogens.

Best regards,

Kasturi Haldar

Editor-in-Chief

PLOS Pathogens

orcid.org/0000-0001-5065-158X

Michael Malim

Editor-in-Chief

PLOS Pathogens

orcid.org/0000-0002-7699-2064